# Lysyl oxidase-like 3 restrains mitochondrial ferroptosis to promote liver cancer chemoresistance by stabilizing dihydroorotate dehydrogenase

Meixiao Zhan [1,4], Yufeng Ding [2,4] ✉, Shanzhou Huang[3,4], Yuhang Liu[2], Jing Xiao[1], Hua Yu [2] ✉, Ligong Lu [1] ✉ & Xiongjun Wang [2] ✉

To overcome chemotherapy resistance, novel strategies sensitizing cancer cells to chemotherapy are required. Here, we screen the lysyl-oxidase (LOX) family to clarify its contribution to chemotherapy resistance in liver cancer. LOXL3 depletion significantly sensitizes liver cancer cells to Oxaliplatin by inducing ferroptosis. Chemotherapy-activated EGFR signaling drives LOXL3 to interact with TOM20, causing it to be hijacked into mitochondria, where LOXL3 lysyl-oxidase activity is reinforced by phosphorylation at S704. Metabolic adenylate kinase 2 (AK2) directly phosphorylates LOXL3-S704. Phosphorylated LOXL3-S704 targets dihydroorotate dehydrogenase (DHODH) and stabilizes it by preventing its ubiquitin-mediated proteasomal degradation. K344-deubiquitinated DHODH accumulates in mitochondria, in turn inhibiting chemotherapy-induced mitochondrial ferroptosis. CRISPR-Cas9-mediated site-mutation of mouse LOXL3-S704 to D704 causes a reduction in lipid peroxidation. Using an advanced liver cancer mouse model, we further reveal that low-dose Oxaliplatin in combination with the DHODH-inhibitor Leflunomide effectively inhibit liver cancer progression by inducing ferroptosis, with increased chemotherapy sensitivity and decreased chemotherapy toxicity.

Chemotherapy for liver cancer has not achieved satisfactory results, and can result in liver damage owing to severe side effects, causing patients to suffer[1,2]. Hence, it is particularly urgent to identify targets that sensitize liver cancer cells to chemotherapy, and to explore a combined strategy to alleviate side effects and drug resistance.

The lysyl-oxidase (LOX) family, comprising LOX and four LOX-like proteins, functions primarily in extracellular matrix (ECM) remodeling and cross-linking of collagen and elastic fibers[3–5]. Aberrant LOX family member expression and activity contributes to a wide range of diseases, including malignant cancers. However, targeting of LOX family members has produced unexpected results[6], indicating that the LOX family might not participate in ECM remodeling. For example, LOXL2 is partially located in the nucleus, and shows histone demethylase activity[7] and histone oxidation activity[8] at H3 lysine 4. We have previously reported[9] that under OSM stimulation, LOXL3 translocates into the nucleus and removes lysine acetylation of STAT3 at K685. Given that the LOX family plays diverse subcellular roles, we wondered which

[1]Zhuhai Interventional Medical Center, Guangdong Provincial Key Laboratory of Tumour Interventional Diagnosis and Treatment, Zhuhai People's Hospital, Zhuhai Hospital affiliated with Jinan University, Zhuhai, 519000 Guangdong, China. [2]Precise Genome Engineering Center, School of Life Sciences, Guangzhou University, 510006 Guangzhou, China. [3]Department of General Surgery, Guangdong Provincial People's Hospital, Guangdong Academy of Medical Sciences, 510080 Guangzhou, China. [4]These authors contributed equally: Meixiao Zhan, Yufeng Ding, Shanzhou Huang. ✉e-mail: yfding@gzhu.edu.cn; yuhua@sibcb.ac.cn; luligong1969@jnu.edu.cn; wangxiongjun@gzhu.edu.cn

its members contribute intracellularly to chemoresistance in liver cancer.

Ferroptosis, a recently discovered mode of regulated cell death characterized by iron-dependent lipid peroxidation accumulation[10], is closely related to progression in diseases including cancer. Ferroptosis is mechanistically and morphologically different from other forms of cell death, such as apoptosis; it therefore has great potential for cancer therapy. Tumor-cell chemoresistance, caused by adaptive tolerance to apoptosis, can be treated by inducing ferroptosis[11]. Hence, it would be interesting to determine whether drugs used in chemotherapy for liver cancer, such as Oxaliplatin, can induce ferroptosis in liver cancer cells. Several ferroptosis-suppressing pathways have been discovered, including the GPX4-mediated reduction of phospholipid hydroperoxides (PLOOHs), ferroptosis suppressor protein 1 (FSP1)–ubiquinone system, and squalene-mediated inhibition of lipid peroxidation[10,12]. We wondered whether the LOX family, comprising extracellular matrix enzymes, participates in regulating ferroptosis in tumor cells.

While the functions of other LOX family members in the liver cancer have been elucidated, those of LOXL3 in liver cancer require further comprehensive research. For example, LOX-overexpression induces epithelial-mesenchymal transition[13]. LOXL2 functions as an oncoprotein in liver cancer, driving liver cancer cell proliferation, migration, invasion, and cell survival[14–16]. LOXL4 promotes liver cancer cell death by activating compromised wild-type p53[17]. LOX family members function both as ECM enzymes and as intracellular proteins that participate in signal transduction or epigenetic regulation[8,9]. We observed, among its members, only LOXL3 was partially localized in the mitochondria, dependent on the translocase of the outer membrane (TOM) complex. The TOM complex, located in the outer mitochondrial membrane, functions primarily as a regular entry gate for mitochondrial proteins[18]. It comprises the receptors TOM20, TOM22, and TOM70, and the channel-forming protein TOM40. TOM20, localized in the outer mitochondrial membrane, achieves initial precursor recognition via a presequence, facilitating protein import across the outer mitochondrial membrane[19].

In this study, LOXL3 was transported into the mitochondria by TOM20, and was phosphorylated by the metabolite kinase AK2. It then exerted stronger lysyl-oxidase activity, preventing DHODH-K344 ubiquitination, in turn conferring resistance to mitochondrial ferroptosis, as mitochondrial DHODH accumulation is required to prevent ferroptosis in cells with low GPX4 expression[20]. To enhance the efficacy of Oxaliplatin and reduce the toxicity and side effects of high-dose treatment, we applied an advanced liver cancer mouse model to explore a combined strategy, low-dose Oxaliplatin with Leflunomide, to efficiently compromise the progression of liver cancer, with reduced drug toxicity.

## Results

### LOXL3 mitochondrial localization and enzyme activity endowed liver cancer cells with tolerance toward chemotherapy-induced ferroptosis

To examine the association between the LOX family and chemoresistance in liver cancer, we first knocked down LOX family members in hepatocellular carcinoma (HCC) cells (Supplementary Fig. 1a), which did not influence survival rate (Supplementary Fig. 1b). Next, when we treated genetically manipulated cells with Oxaliplatin and 5-Fu, LOXL3 depletion greatly sensitized liver cancer cells to these drugs, whereas depletion of other members of the LOX family had no significant effects on chemotherapy resistance (Fig. 1a, Supplementary Fig. 1c). Consistent with this, LOXL3-deficient cells exhibited elevated cell death and lower IC50 value which represented the lower chemoresistance status of cancer cells under the drug treatment (Supplementary Fig. 1d, e). Meanwhile, we further confirmed the function of LOXL3 on

chemotherapy resistance by using two shRNAs targeting LOXL3 to avoid off-targeting effect (Supplementary Fig. 1f–h). Moreover, we also excluded the possibility that LOXL3 depletion affected the expression of other LOX members (Supplementary Fig. 1f). LOXL3-deficient cells rescued with enzyme-dead (ED)-LOXL3 showed reduced cell viability in response to chemotherapy, proving that LOXL3 lysyl-oxidase activity is involved in chemoresistance (Fig. 1b, Supplementary Fig. 1i–k).

In cancer cells, most chemotherapeutics generate robust production of reactive oxygen species (ROS), which play important roles in regulating cell death and drug resistance[21]. Here, Oxaliplatin treatment induced robust ROS production in liver cancer cells. While LOXL3 deficiency only slightly altered cytosolic ROS levels (Fig. 1c, Supplementary Fig. 1j), lipid peroxidation was substantially elevated LOXL3 under Oxaliplatin treatment in LOXL3-deficient cells (Fig. 1d, Supplementary Fig. 1k). As ferroptosis is recognized as a new form of cell death featured by severe iron-dependent lipid peroxidation relying on ROS generation, we further evaluated if LOXL3-deficient-cell regulated ferroptosis in response to Oxaliplatin treatment. Via transmission electron microscopy, the feature of ferroptosis, such as dense mitochondrial structure with significantly reduced mitochondrial ridges, and the smaller volume of mitochondrial, was observed (Fig. 1e). Additionally, for further conformation, we added the ferroptosis-inhibitor ferrostatin-1 which blocked Oxaliplatin-induced ferroptosis in LOXL3-depleted cells (Supplementary Fig. 1l, m). Notably, the classic ferroptosis inducer Erastin phenocopied Oxaliplatin-induced ferroptosis (Supplementary Fig. 1n, o).

For the aim to increase chemosensitivity and reduce the toxicity triggered by chemotherapeutic drugs, we introduced the low-dose treatment into our study to investigate whether LOXL3 deficiency still sensitizes HCC cells to the low-dose Oxaliplatin treatment by inducing ferroptosis, which was confirmed by measuring lipid peroxidation (Supplementary Fig. 1p). Next, for the validation of chemotherapy initiated ferroptosis induced by which one pathway, we detected five important ferroptosis-related markers and one apoptosis marker, caspase 3 and its cleavage form. Only DHODH protein exhibited reduced levels in LOXL3-deficient or ED-LOXL3 cells (Fig. 1f). Consistent with this, LOXL3-deficient or ED-LOXL3 cells were more sensitive than wild-type (WT) -LOXL3-expresssing cells to normal or low-dose Oxaliplatin (Fig. 1g, Supplementary Fig. 1q).

Next, fractionation of the cytoplasm, nucleus, and mitochondria revealed that LOXL3 was partially distributed in the mitochondria, and its mitochondrial localization was reinforced by chemotherapy (Fig. 1h, i), while LOX and LOXL2 in the nucleus, consistent with previous reports[22,23]. We constructed LOXL3-deficient cells which were reintroduced expression of WT or ED-LOXL3 with a mitochondrial signal peptide (hereafter, M-WT-L3 and M-ED-L3, respectively) (Supplementary Fig. 1r). Fractionation assays confirmed the mitochondrial localization of these cells (Supplementary Fig. 1s). Cells expressing WT-L3 or M-WT-L3 effectively suppressed the strong increase in lipid peroxidation in LOXL3-deficient cells under Oxaliplatin treatment, but not in the ED-forms (Fig. 1j, k), which implied the dependence of LOXL3 enzyme activity in mitochondrial. Correspondingly, cell viability and cell death were measured (Fig. 1l, Supplementary Fig. 1t). To ascertain the relative mitochondrial lysyl-oxidase activity of LOXL3, we forced the expression of FLAG-tagged WT-LOXL3 into HCC cells and enriched the mitochondrial or cytosolic LOXL3 protein to measure the LOXL3 activity. It was observed that mitochondrial LOXL3 exhibited higher activity than cytosolic LOXL3 while LOXL3 activity was upregulated by Oxaliplatin (Supplementary Fig. 1u). Additionally, we next forced the expression of FLAG-tagged WT-LOXL3 and M-WT-L3 (Supplementary Fig. 1v). M-LOXL3 just showed a little stronger activity than total LOXL3 (Supplementary Fig. 1w), implied that higher LOXL3 activity in mitochondria response to Oxaliplatin was synergistically controlled by the upstream signaling and the intensity of LOXL3 in mitochondria.

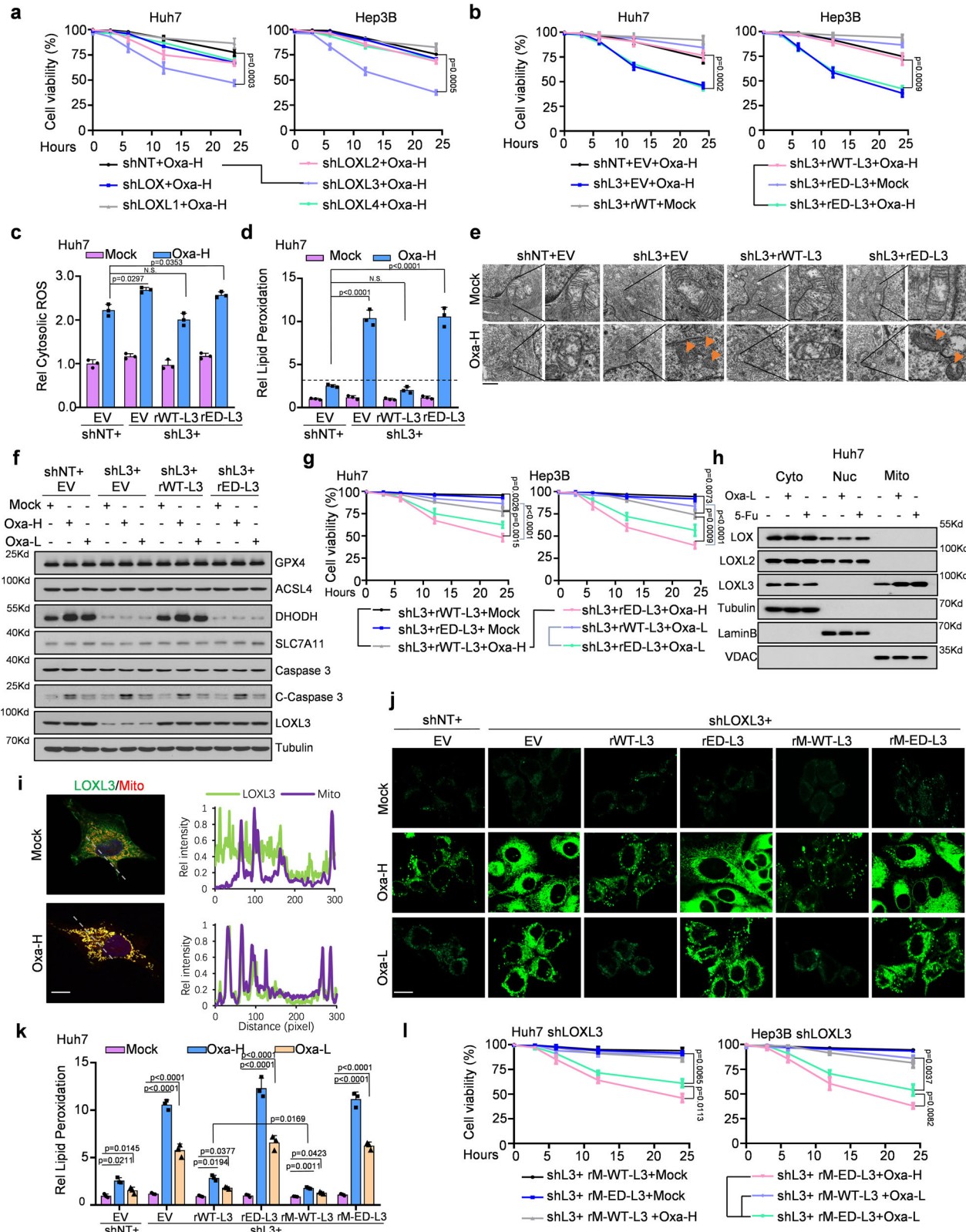

In summary, LOXL3 was partially located in the mitochondria, exhibiting increased entry into the mitochondria and enzyme activity under drug treatment. This could help cancer cells to resist chemotherapy-induced lipid peroxidation and ferroptosis, possibly via the LOXL3/DHODH axis. Importantly, LOXL3 deficiency sensitized liver cancer cells to low-dose Oxaliplatin treatment, with reduced toxicity and increased tumor-killing efficacy.

**EGF−EGFR signaling mediated the LOXL3−TOM20 axis, blocking chemotherapy-induced ferroptosis, and upregulating LOXL3 phosphorylation at S704**

We performed cytokine screening to elucidate the signaling involved in mediating LOXL3 translocation into mitochondria: EGF treatment promoted LOXL3 entry into mitochondria; TNFα treatment induced a substantial reduction in cytosolic LOXL3, but did not increase

**Fig. 1 | LOXL3 mitochondrial location and enzyme activity endow liver cancer cells with tolerance to chemotherapy-induced ferroptosis. a** Cell viability of Huh7 or Hep3B cells stably expressing shNT, shLOX, shL1, shL2, shL3, or shL4, treated with Oxa-H for the indicated time points. **b**–**d** Huh7 or Hep3B cells stably expressing shL3 with restored expression of EV, WT- or ED-L3 were treated with Oxa-H for the indicated time points, and cell viability was assessed (**b**). Cytosolic ROS levels of Huh7 cells were assessed by fluorescence intensity at $E_x/E_m$=495/529 nm (**c**) and lipid peroxidation levels of Huh7 cells were assessed by flow cytometry using BODIPY C11(**d**). **e**–**g** Huh7 or Hep3B cells stably expressing shL3 with restored expression of EV, WT- or ED-L3 were treated with Oxa-H or Oxa-L for 6 h (**e**) or 24h (**f**) or indicated time points (**g**). Above indicated Huh7 cells treated with Oxa-H were executed with post-embedding staining and imaged by transmission electron microscopy using a gold labeled protein method to obtain ultrastructural evidence (**e**). Western blot (WB) was performed using indicated antibodies (**f**). Cell viability was assessed. (**g**). Scale bars: 1 μm. **h**-**i** Huh7 cells were treated with Oxa or 5-Fu for half hour and then the mitochondrial, cytosolic and nuclear fractions were collected for WB (**h**) using indicated antibodies or IF(**i**) using FLAG antibody and MitoTracker. Scale bars: 20 μm. **j**–**l** Indicated cells were treated with Oxa-H/L and collected for lipid peroxidation level detection were assessed and calculated by fluorescence microscopy or flow cytometry using BODIPY C11 (**j**, **k**), or cell viability measurement (**l**). Scale bars: 50 μm. For convenience, LOXL protein family were named as L, such as L1, L2, L3 or L4. Oxa-H, high-dose Oxaliplatin, 5 μg ml$^{-1}$; Oxa-L, low-dose Oxaliplatin, 1 μg ml$^{-1}$. ED, enzyme dead, H607/609Q. EV, empty vector. For (**a**–**d**, **g**, **k**, **l**), data present means ± SEM from three independent experiments ($n$ = 3). The statistical analysis was calculated by two-way ANOVA for multiple comparisons (**a**, **b**, **g**, **l**) or one-way ANOVA with Tukey's honest significance difference (HSD) post hoc test (**c**, **d**, **k**). Source data are provided as a Source Data file. See also Supplementary Fig. 1.

mitochondrial LOXL3 (Fig. 2a). IF-staining revealed that EGF drive LOXL3 entry into mitochondria (Fig. 2b).

To explore the role of EGFR signaling in chemotherapy-induced resistance to ferroptosis, we combined low-dose Oxaliplatin and the EGFR monoclonal antibody Cetuximab to treat cancer cells. Relative to Oxaliplatin or Cetuximab alone, this combination significantly increased lipid peroxidation and cell death in WT-LOXL3-expressing cells, but not in LOXL3-deficient cells (Supplementary Fig. 2a, b). In LOXL3-deficient cells, Oxaliplatin was more effective in combination with Lenvatinib than with Cetuximab. Relative to the combination of Oxaliplatin and Lenvatinib, combining all three (Oxaliplatin, Cetuximab, and Lenvatinib) increased lipid peroxidation in WT-LOXL3-expressing cells, but not in LOXL3-deficient cells (Supplementary Fig. 2a, b). These results reveal that EGFR signaling mediated inhibition of lipid peroxidation depends on LOXL3; further Lenvatinib improves the efficiency of EGFR inhibition or LOXL3-depletion in chemotherapy-induced ferroptosis.

Since TOM20 usually acts as a general mitochondrial import receptor[24], co-immunoprecipitation (Co-IP) assay confirmed LOXL3 was observed in the TOM complex, mediated by EGF treatment, implying that EGF signaling may mediate the LOXL3–TOM20 axis and mitochondrial localization of LOXL3 (Supplementary Fig. 2c, Fig. 2c). AlphaFold 2 showed a presequence with amino acids 1–40 in the N-terminus of LOXL3, forms a short helix rich in positively charged amino acids. Hence, we suspected that the salient sequence of LOXL3 could act as a hook for recruiting TOM20 (Fig. 2d). To test this, we deleted this sequence, and observed that the association between LOXL3 with TOM20 declined substantially (Fig. 2e). Mutating K35/36 to A35/36 impaired the interaction between LOXL3 and TOM20, as expected (Fig. 2f). Consequently, K35/36A-LOXL3 had much weaker mitochondrial localization than WT-LOXL3 (Fig. 2g, Supplementary Fig. 2d, e). Correspondingly, LOXL3-knockdown cells with restored K35/36A-LOXL3 expression exhibited more lipid peroxidation than those with restored WT-LOXL3 expression (Fig. 2h, i, Supplementary Fig. 2f). Further, LOXL3-knockdown cells with restored K35/36A-LOXL3 expression exhibited reduced DHODH expression (Fig. 2j). Meanwhile, to test whether the canonical function of LOXL3 existed in the chemoresistance regulation of HCC, we restored the expression of K35/36A-LOXL3 or WT-LOXL3 in LOXL3-deficient cells. Since LOXL3-K35/K36A has the major canonical activity of LOXL3 in the secreted medium (Supplementary Fig. 2g, h), but presents a lack of the ability to enter into mitochondrial (Fig. 2f, g, Supplementary Fig. 2d, e), thus, the restored expression of LOXL3-K35/K36A in LOXL3-knockdown cells could ascertain the role of LOXL3 in HCC chemoresistance regulation through the canonical function or not. We observed that LOXL3-WT, but not LOXL3-K35/K36A mutant, could rescue the lipid peroxidation level (Fig. 2h, i), cell death level, drug response status (Supplementary Fig. 2i, j) and DHODH protein level (Fig. 2j) under the treatment of Oxaliplatin, which eventually excluded the canonical function of LOXL3 in chemoresistance of liver cancer.

In order to find the upstream in mitochondria how to regulate the LOXL3, we compared the post-translational modification of LOXL3, via mass spectrometry, in LOXL3-deficient cells with restored K35/36A-LOXL3 or WT-LOXL3 expression. Based on the heatmap, mitochondrial LOXL3-S704 phosphorylation was significantly upregulated, whereas that of S39 and S563 was slightly different between cytosolic and mitochondrial LOXL3 (Fig. 2k, Supplementary Fig. 2k). In the lysyl-oxidase activity test, we mimicked dephosphorylation by mutating S704 to A704: LOXL3 activity decreased by >about 65%, while dephosphorylation mimicry of S39 and S563 did not significantly affect LOXL3 activity (Fig. 2l). A specific antibody against LOXL3-S704 phosphorylation was then prepared (Supplementary Fig. 2l, m) and showed S704 phosphorylation occurred only in mitochondrial LOXL3 (Fig. 2m). Of note, the blockade of EGFR signaling by Cetuximab would diminish Oxaliplatin induced upregulation of LOXL3-S704 phosphorylation (Supplementary Fig. S2n), further proving that chemotherapy induced EGFR activation is required for mitochondrial LOXL3-S704 phosphorylation mediated anti-ferroptosis.

Together, these findings reveal that EGF–EGFR signaling mediates the interaction between LOXL3 and TOM20, leading to mitochondrial LOXL3-S704 phosphorylation. This in turn increases LOXL3 activity, reducing chemotherapy-induced lipid peroxidation and ferroptosis.

## LOXL3 phosphorylation prevents DHODH-K344 ubiquitination and promotes DHODH stability to resist chemotherapy-induced ferroptosis

We have previously reported that LOXL3 activity is increased when it forms homodimer[9]. We tested whether the S704A-LOXL3 mutant, which mimics dephosphorylation, reduces homodimerization. Co-IP and reverse high-performance liquid chromatography (HPLC) showed that the S704A mutant almost completely eliminated LOXL3 homodimerization (Supplementary Fig. 3a, Fig. 3a). The DHODH protein level, but not its mRNA level, was downregulated in LOXL3-knockdown cells. Restoring the expression of WT-LOXL3, but not of S704A-LOXL3, rescued the DHODH protein level, implying that S704 phosphorylation of LOXL3 is required for DHODH protein stability (Fig. 3b, c, Supplementary Fig. 3b, c). The fact that LOXL3-S704 phosphorylation mediated the association between LOXL3 and DHODH was confirmed via Co-IP (Fig. 3d). Colocalization of LOXL3 and DHODH was disrupted in cells expressing the LOXL3-S704A mutant (Supplementary Fig. 3d).

To further confirm that LOXL3-S704 phosphorylation regulates DHODH protein stability, we treated cells with cycloheximide (CHX) and observed the degradation rate of DHODH protein was substantially accelerated in LOXL3-knockdown cells and in the restored S704A-LOXL3 mutant cells (Fig. 3e, Supplementary Fig. 3e). The most ubiquitinated DHODH sites in the two cell lines expressing restored WT or S704A mutated LOXL3 were compared and summarized in a heatmap (Fig. 3f). These lysine sites were mutated one-by-one to arginine, to test whether this would rescue the DHODH protein level (Fig. 3g). DHODH ubiquitination was significantly reduced after

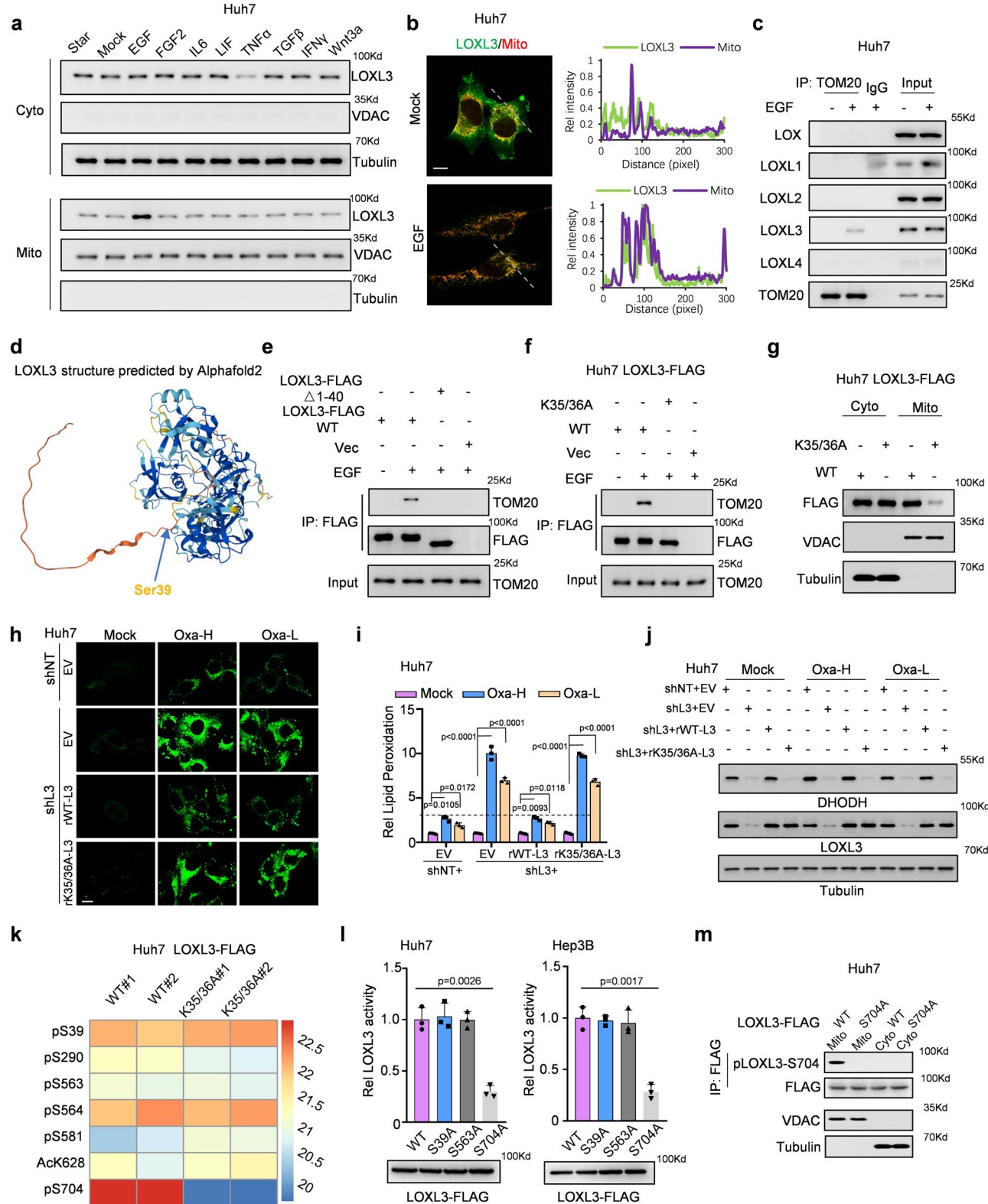

mutating K344 to R344 (Fig. 3h), consistent with the most pronounced change in the DHODH-K344 ubiquitination signal (Fig. 3f). Relative to levels in the WT-LOXL3-overexpressing cells, DHODH-K344 oxidation was lower in cells overexpressing S704A-LOXL3, but higher in those expressing S704D-LOXL3 (Fig. 3i), indicating that LOXL3-S704 phosphorylation increased LOXL3 lysyl-oxidase activity and promoted DHODH-K344 oxidation. Combined with our prior observation that LOXL3 interacts with DHODH and regulates its protein stability, we can

conclude that LOXL3 catalyzes DHODH-K344 oxidation via its lysyl-oxidase activity, and competes with DHODH-K344 ubiquitination to prevent degradation; its competitiveness is strengthened by LOXL3-S704 phosphorylation in the mitochondria.

Under the chemotherapy (Oxaliplatin) treatment, deubiquitinated and stable DHODH (K344R) reduced the upregulated level of lipid peroxidation triggered by unphosphorylated LOXL3 (S704A) (Fig. 3j), verifying that DHODH acts as a downstream effector of LOXL3

**Fig. 2 | EGF/EGFR signaling mediated the LOXL3/TOM20 axis, which blocked chemo-induced ferroptosis and upregulated LOXL3 phosphorylation at S704.** **a** Huh7 cells were starved for over 12 h and then treated with cytokines for half an hour. Then, the mitochondrial and cytosolic fractions were collected for WB. **b**, **c** Huh7 cells were starved for over 12 h, treated with EGF for half an hour, then fixed and stained for IF (**b**) or mitochondrial proteins were extracted for IP with an antibody against TOM20(**c**). Scale bars: 20 μm. **d** AlphaFold 2 predicted the LOXL3 structure conformation. The arrow indicates Ser39 in the N-terminus (amino acids 1-40) of LOXL3, which contains a helix. **e**, **f** Huh7 cells stably expressing WT- or 1-40 amino acid-deleted LOXL3-FLAG(**e**), or K35/36A-mutant LOXL3-FLAG (**f**) were starved for over 12 h and then treated with EGF for half an hour. Then, the cells were collected for Co-IP. **g** Huh7 cells stably expressing WT- or K35/36A-mutant LOXL3-FLAG were collected to extract total protein for fractionation of mitochondrial and cytosol. **h–j** Huh7 cells expressing shNT with EV or expressing shL3 with EV, WT- or K35/36A-mutant LOXL3 were collected for lipid peroxidation measurement (**h**). Respective lipid peroxidation levels were assessed and calculated by flow cytometry using BODIPY C11 (**i**). The same batch cells were collected and WB was performed using the indicated antibodies (**j**). Scale bars: 50 μm. **k** Huh7 cells expressing WT- or K35/36A-mutant LOXL3 were collected, and enrichment of LOXL3 was identified by LC/MS analysis of posttranslational modifications. The sites with differential levels of posttranslational modifications were organized as a heatmap using two batches of data. **l** Huh7 or Hep3B cells expressing WT- or mutant LOXL3-FLAG were enriched for LOXL3 lysyl-oxidase activity. **m** Huh7 cells expressing WT- or S704A-mutant LOXL3-FLAG were collected to fraction the mitochondria and cytosol. For (**i**, **l**), data present means ± SEM from three independent experiments ($n = 3$) and the statistical analysis was calculated by one-way ANOVA with Tukey's HSD post hoc test (**i**, **l**). Source data are provided as a Source Data file. See also Supplementary Fig. 2.

under chemotherapy. Consistent with this, K344R-DHODH rescued cell viability (Fig. 3k) and inhibited the cell death of S704A-LOXL3-expressing cells, even with low-dose Oxaliplatin treatment (Supplementary Fig. 3f, g). Notably, IF staining revealed that low GPX4 expression was associated with minimal mitochondrial localization (Supplementary Fig. 3h), indicating that mitochondrial ferroptosis in Huh7 and Hep3B cells was determined by the accumulation of mitochondrial DHODH.

DHODH catalyzes the conversion of Dihydroorotate (DHO) to orotic acid (OA) in the pyrimidine biosynthesis pathway. In the catalytic process, DHODH removes two electrons from DHO to mitochondrial respiratory complex via CoQ, leading to the reduction of CoQ to $CoQH_2$ in the mitochondrial inner membrane which resulted in the tight combination of the effect of DHODH on mitochondrial respiratory chain with ferroptosis regulation[20,25]. Firstly, we investigated whether the mitochondrial redox function of DHODH, or its pyrimidine synthesis capacity, contributes to the fight against Oxaliplatin-induced ferroptosis. Uridine, a DHODH metabolite, was markedly downregulated in LOXL3-S704A cells. However, the supplementation with uridine did not enhance Oxaliplatin resistance in LOXL3-S704A cells, indicating that pyrimidine synthesis by DHODH did not participate in LOXL3-mediated resistance to ferroptosis (Supplementary Fig. 3i–k). To confirm the idea that DHODH mediated the effect of LOXL3 on ferroptosis by reducing CoQ to $CoQH_2$, we supplemented mitochondria-targeted analogues of CoQ and $CoQH_2$ to LOXL3-S704A mutant cells under Oxaliplatin treatment. The result showed that $CoQH_2$, but not CoQ, could protect the lipid peroxidation and cell death induced by DHODH deficiency in LOXL3-S704A mutant cells under oxaliplatin treatment (Supplementary Fig. 3l, m). As electron transport chain (ETC) complex III converts $CoQH_2$ back into CoQ, we treated cells with the complex III inhibitor antimycin A (Anti A) to decrease the $CoQ/CoQH_2$ ratio. Similar with the mitoQH_2 supplementation, antimycin A protected cells against Oxaliplatin treatment-induced ferroptosis and cell death (Supplementary Fig. 3n, o). Although previous reports showed that no more than 10% of routine oxygen consumption and ATP production within OXPHOS in DHODH deficiency cells and DHODH knockout cells feature normal levels of OXPHOS-derived ATP and bioenergetics[26], we measured the level of overall oxygen consumption rate and ATP production in LOXL3-S704 mutant cells with downregulated DHODH expression. The result indeed confirmed the slight decrease (Supplementary Fig. 3p, q), which may be explained that CoQ serves as an electron acceptor both for DHODH and the other upstream ETC complexes specific in liver cancer cells, leading to the overall balance of mitochondrial respiratory chain.

These findings indicate that LOXL3-S704 phosphorylation targets and prevents DHODH-K344 ubiquitination, in turn promoting DHODH stability to resist ferroptosis by reducing CoQ to $CoQH_2$ in mitochondria and upregulate cell survival after Oxaliplatin treatment resulting in the increased chemoresistance.

## Mitochondrial adenylate kinase 2 phosphorylates LOXL3-S704 and confers resistance to chemotherapy-induced ferroptosis

We screened for the kinases involved in mitochondrial localization and identified the kinase responsible for LOXL3-S704 phosphorylation, via a dot blot screen (Fig. 4a). AK2 is present in the upper right corner, indicating that it is the most prominent kinase regulating LOXL3-S704 phosphorylation (Fig. 4b). Western blot and dot blot analyses revealed that AK2-knockdown greatly reduced LOXL3-S704 phosphorylation (Fig. 4c, Supplementary Fig. 4a). AK2 depletion substantially impaired LOXL4-S704 phosphorylation, even upon EGF stimulation (Fig. 4d). Co-IP confirmed that LOXL3 and AK2 may form an axis independent of AK2 activity (Fig. 4e, Supplementary Fig. 4b). AK2 depletion or inactivation by an AK2 inhibitor resulted in robust upregulation of lipid peroxidation, consistent with the results from cells expressing activity-deficient LOXL3 (Fig. 4f, Supplementary Fig. 4c). AK2 enzyme activity was positively correlated with LOXL3-S704 phosphorylation, revealed via forced expression of WT- or ED-AK2 (Fig. 4g, h).

Next, to further investigate how AK2 participates in regulating LOXL3, we constructed endogenous AK2-knockdown cells and restored the expression of WT-AK2 or mitochondrial signal peptide mutant-AK2 (K14G/R17G) (Fig. 4i). IF staining confirmed that this mutant AK2 did not enter the mitochondria (Supplementary Fig. 4d). In cells expressing mutant AK2, lipid peroxidation levels and sensitivity to Oxaliplatin were significantly elevated (Fig. 4J), and LOXL3-S704 phosphorylation was reduced (Supplementary Fig. 4e), probably because cytosolic AK2 did not efficiently form an axis with LOXL3, or because of its lower kinase activity in the cytosol (Supplementary Fig. 4f). The restored expression of S704D-LOXL3 in AK2-deficient cells efficiently rescued the resistance of tumor cells to chemotherapy, further validating that LOXL3 acts downstream of AK2 (Fig. 4k).

These findings indicate that LOXL3-S704 phosphorylation occurs in the mitochondria and is mediated by mitochondrial kinase AK2. Although AK2 is a well-known mitochondrial kinase, we demonstrated that LOXL3 is a protein substrate of AK2.

## S704D-Loxl3 confers resistance to Oxaliplatin treatment in sleeping beauty transposon-induced liver cancer

To determine the significance of clinical treatment for HCC patients with high LOXL3 activity and more chemotherapy resistance, we constructed S704D-Loxl3 mice, and confirmed their construction by sequencing (Fig. 5a, b, Supplementary Fig. 5a). We applied sleeping beauty (SB) transposon-mediated expression of activated β-catenin and c-Met to promptly induce advanced mouse HCC, then administered Oxaliplatin treatment (Fig. 5c). Compared to the WT mice, there was a moderate difference in liver weight after SB induction (Supplementary Fig. 5b). After receiving Oxaliplatin treatment, the S704D-Loxl3 mice still exhibited abdominal distention, and liver dissection revealed elevated liver volume and weight (Fig. 5d). H&E staining revealed that marked accumulation of tumor tissue in the livers of S704D-Loxl3 mice (Fig. 5e). ALT and AST were upregulated in the

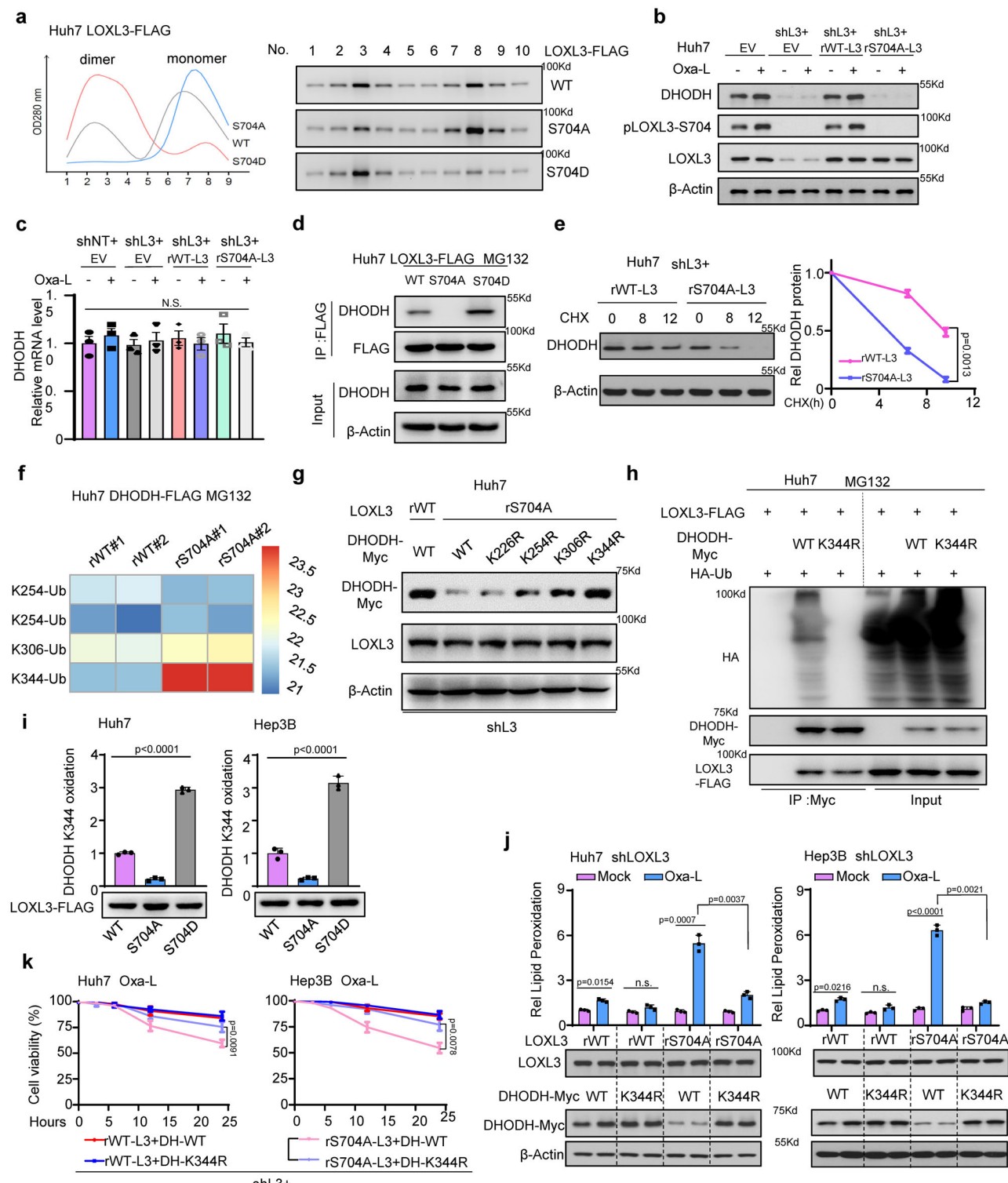

S704D-mice livers, providing a preliminarily indication of abnormal liver function (Fig. 5f, Supplementary Fig. 5c). Following Oxaliplatin treatment, the livers of *S704D-Loxl3* mice exhibited synchronous upregulation of DHODH levels and activity (Fig. 5g, Supplementary Fig. 5d), with significant downregulation of lipid peroxidation LOXL3, to about 20% of that in the livers of wild-type mice (Fig. 5h). This indicates that there was minimal Oxaliplatin-induced ferroptosis in *S704D-Loxl3* mice, leading to the significant resistance to Oxaliplatin treatment for SB-induced advanced liver cancer, showed by liver weight ratio and the average number of tumors in each liver, and

reduced survival time (Fig. 5i–k). These findings indicate that *S704D-Loxl3* mice exhibited blockage of ferroptosis and reduced lipid peroxidation, thereby promoting chemotherapy resistance.

## Low-dose Oxaliplatin with LOXL3−DHODH axis inhibition efficiently facilitates treatment in advanced liver cancer

To explore the treatment strategy of utilizing low-dose Oxaliplatin to dampen the LOXL3−DHODH axis, we first performed subcutaneous transplantation of tumor cells expressing the S704A mutant, or of control tumor cells: there was no significant difference in solid-tumor

**Fig. 3 | LOXL3-S704 phosphorylation prevents DHODH-K344 ubiquitination and promotes DHODH stability to resist chemotherapy-induced ferroptosis.** **a** Huh7 cells stably expressing WT- or S704A-, S704D-mutant LOXL3-FLAG were enriched for R-FPLC analysis. **b, c** Indicated Huh7 cells with restored expression of WT- or S704A-mutant LOXL3 were treated with Oxa-L for WB (**b**) or qRT-PCR(**c**). **d** Huh7 cells expressing WT-, S704A- or S704D-mutant LOXL3-FLAG were treated with MG132 for 12 h and collected for Co-IP. **e** Huh7 shL3 cells expressing WT- or S704A-mutant LOXL3 were treated with CHX (2 µM) for the indicated time points and collected for WB (**e**, left). The protein decay was analyzed (**e**, right). CHX: cycloheximide. **f, g** Huh7 shL3 cells expressing WT or S704A mutant LOXL3 were transfected with DHODH-FLAG or DHODH-myc, including WT and mutants with the above lysine (K) sites mutated to arginine (R). Then, WT-DHODH was enriched and subjected to identify ubiquitin modifications by LC/MS analysis after cells were treated with MG132 for 12 h (**f**). The WT and KR mutants DHODH-myc protein level was detected by WB (**g**). **h** Huh7 cells expressing WT- or S704D-LOXL3 were transfected with DHODH-myc and HA-ubiquitin for 36 h and treated with MG132 for 12 h. An IP assay was performed and WB was carried out using the indicated antibodies. **i** Huh7 or Hep3B cells expressing WT- or S704A-, S704D-mutant LOXL3-FLAG were collected to enrich LOXL3-FLAG, which was incubated with peptide "ALEK$_{344}$IRAGAS" as LOXL3 substrate. **j** Huh7 or Hep3B shL3 cells expressing WT- or S704A-mutant LOXL3 were transfected with WT- or K344R-mutant DHODH-FLAG for 48 h, and then lipid peroxidation levels and WB were assessed. **k** Huh7 or Hep3B shL3 cells expressing restored WT- or S704A-mutant LOXL3 with WT or K344R-mutant DHODH were treated with Oxa-L for the indicated time points, and cell viability was assessed. L3, LOXL3; DH, DHODH. For (**c, e, i, k**), data present means ± SEM from three independent experiments ($n = 3$) and the statistical analysis was calculated by one-way ANOVA with Tukey's HSD post hoc test (**c, i, j**) or two-way ANOVA for multiple comparisons (**e, k**). Source data are provided as a Source Data file. See also Supplementary Fig. 3.

formation (Fig. 6a). Under low-dose Oxaliplatin treatment, solid tumors formed by S704A-LOXL3 cells gradually shrank, whereas WT-LOXL3 cells grew rapidly, by recovering from the initial inhibition of proliferation (Fig. 6b). Ki67 staining confirmed that the proliferation rates of WT and S704A tumor cells were relatively similar (Fig. 6c), regardless of the chemotherapy strategy used. However, TUNEL staining revealed that under Oxaliplatin treatment, there were more TUNEL-positive cells in the solid tumors formed by S704A cells than in the control cells (Fig. 6d). This may be because S704A cells are sensitive to Oxaliplatin-induced ferroptosis, as indicated by increased lipid peroxidation (Fig. 6e).

To further explore the therapeutic strategies, we dissected the livers of experimental mice receiving the combination strategy (Fig. 6f), revealing lower liver volume and weight following the combination treatment (Fig. 6g, Supplementary Fig. 6a). H&E staining revealed that the combination strategy compromised liver cancer progression (Fig. 6h). DHODH activity in the liver was diminished by Leflunomide, but increased by low-dose Oxaliplatin treatment (Fig. 6i).

*Loxl3-S704D* mutant mice with higher LOXL3 activity, potentially resistant to Oxaliplatin treatment. WT and *Loxl3-S704D* mutant mice were treated with Oxaliplatin combined with or without Leflunomide (a DHODH inhibitor). Overall, the results showed combined Leflunomide and Oxaliplatin, but not Oxaliplatin alone, efficiently dampened or blocked mouse liver tumor growth with *Loxl3-S704D* mutation (Supplementary Fig. 6b–f). Additionally, combined treatment using Leflunomide and low-dose Oxaliplatin constrained tumor mass and number (Fig. 6j, k), achieving a much healthier liver and prolonged survival time (Fig. 6l). These findings reveal the efficacy of combination low-dose Oxaliplatin and Leflunomide in treating advanced mouse liver cancer.

## The AK2−LOXL3−DHODH axis predicts prognosis, providing a combination strategy for advanced liver cancer

To further investigate the clinical relevance of AK2-mediated phosphorylation of LOXL3 and DHODH proteins in liver cancer, we first mined the TCGA-LIHC cohort and showed LOXL3 mRNA was upregulated in tumor tissues (Supplementary Fig. 7a). Then, we collected samples from 60 patients receiving the FOLFOX chemotherapy treatment, including Oxaliplatin plus Fluorouracil. Of these, 25 patients had paired clear paracancerous or normal tissue. We first detected LOXL3-S704 phosphorylation, and AK2 and DHODH protein levels, in the cancer and adjacent tissues of these 50 HCC patients, using the indicated antibodies with the first validating their specificity (Supplementary Fig. 7b): LOXL3-S704 phosphorylation was significantly upregulated in the human HCC tissue (Fig. 7a).

After pathological grading of the human HCC samples from these 60 patients, we graded the intrahepatic metastasis level as low or high: LOXL3-S704 phosphorylation was significantly upregulated in the high-grade tissues (Supplementary Fig. 7c). DHODH protein levels,

revealed via IHC, were consistent with LOXL3-S704 phosphorylation (Supplementary Fig. 7d).

To precisely calculate the correlation between AK2, LOXL3-S704 phosphorylation, and DHODH, we stained the tumor tissues of the 60 HCC patients using antibodies against AK2 while pLOXL3-S704 and DHODH were measured previously (Supplementary Fig. 7c, d). The staining intensity revealed positive correlations between AK2 and pLOXL3-S704 and between pLOXL3-S704 and DHODH (Fig. 7b, c). The 60 samples were divided into high and low staining-intensity groups, according to their AK2, pLOXL3-S704, and DHODH staining scores. The overall survival curve reveals that the higher levels of AK2, pLOXL3-S704, and DHODH staining predicted worse prognosis (Fig. 7d). Additionally, the progression free survival (PFS) analysis of the patients receiving chemotherapy in the cohort, showed AK2−LOXL3−DHODH axis should confer resistance for HCC patients to chemotherapy (Supplementary Fig. 7e). And the oncogenic role of LOXL3 in liver cancer was further supported by TCGA-LIHC cohort (Supplementary Fig. 7f).

In low-dose Oxaliplatin treatment test using patient-derived xenograft (PDX) models, the group with lower levels of both pLOXL3-S704 and DHODH showed significant sensitivity to Oxaliplatin (Fig. 7e), indicating an appropriate context for human HCC patients was required for selecting Oxaliplatin treatment. Consistent with this phenotype, lower accumulation of DHODH and higher lipid peroxidation under Oxaliplatin treatment were observed in the PDX tumor expressing lower levels of both pLOXL3-S704 and DHODH (Supplementary Fig. 7g, h). In summary, we applied Leflunomide, a DHODH inhibitor, to block AK2−LOXL3−DHODH axis transduction, and to enhance the sensitivity of liver cancer cells to Oxaliplatin treatment by increasing ferroptosis (Fig. 7f). This offers a potentially promising strategy to block advanced HCC with highly activated LOXL3.

## Discussion

Tumor cells, and particularly liver cancer cells, respond to chemotherapeutics such as Oxaliplatin in a way that enables them to survive, grow and proliferate. Tumor cell chemoresistance, caused by adaptive tolerance to apoptosis, can be treated by inducing ferroptosis[11]. Hence, it's necessary to explore a novel type of cell death provides an efficient strategy to overcome chemoresistance and reduce toxic side effect. In this study, we revealed that LOXL3 participates in chemotherapy-induced ferroptosis by activating the AK2−LOXL3−DHODH axis in the mitochondria. Additionally, we explored a combination strategy to improve Oxaliplatin efficacy in advanced liver cancer. Although this study demonstrates the molecular mechanisms and clinical significance of mitochondrial LOXL3, several factors, including study limitations, remain to be discussed.

To explore the association between the LOX family and chemotherapy, we depleted LOX family members individually in these two cell lines and observed LOXL3 depletion sensitized liver cancer cells to

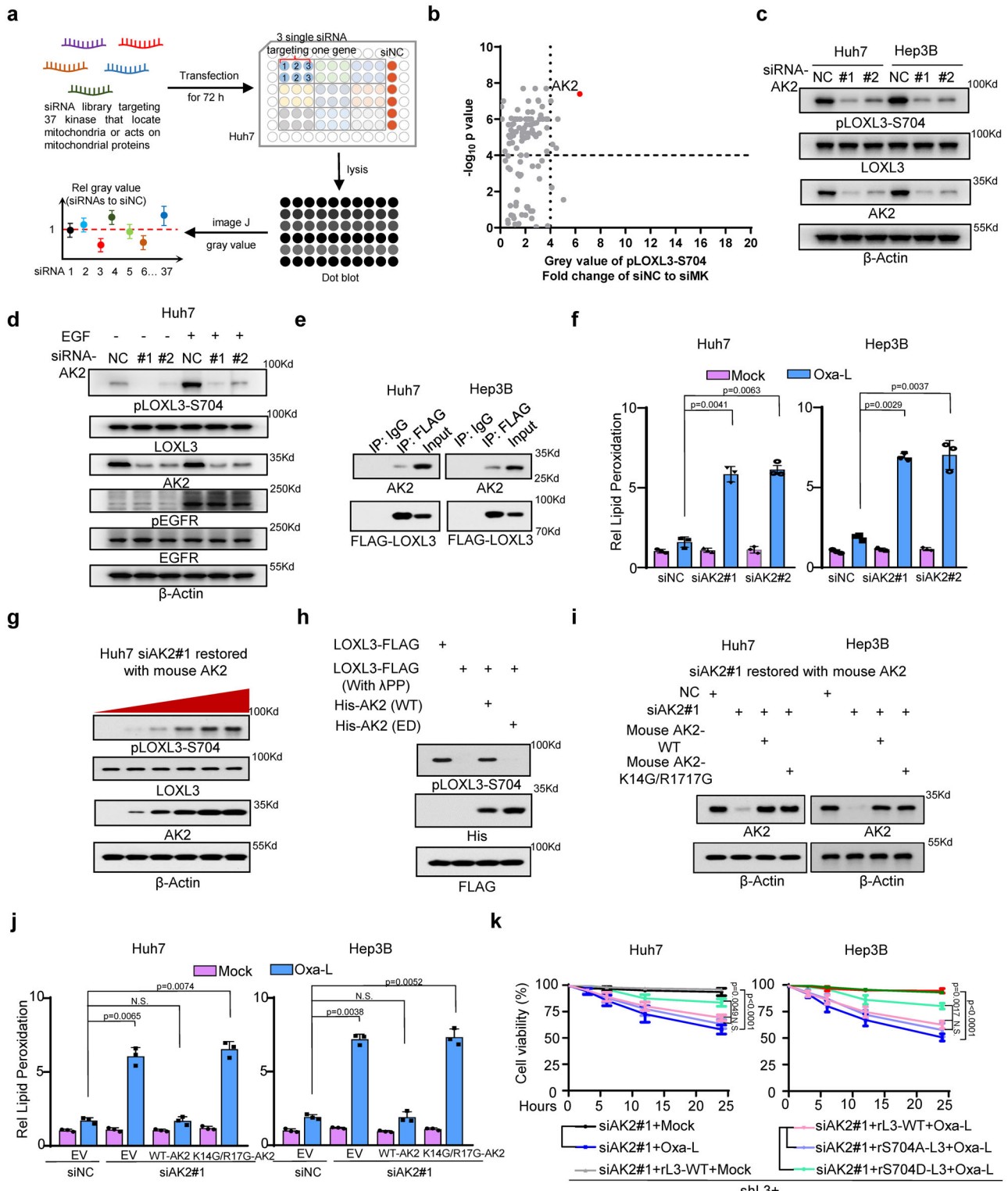

chemotherapeutic drugs. Targeting the LOX family is an exciting prospect for the development of new drugs to prevent cancer progression and metastasis. Preclinical studies have examined the targeting of LOX or LOXL2 by small irreversible competitive inhibitors, and the use of specific function-blocking antibodies to prevent metastasis[6,27]. However, the role of LOXL3 in cancer has not been fully elucidated; thus, inhibitors or monoclonal antibodies that specifically target LOXL3 have not been developed.

Unexpectedly, we observed that mitochondrial lipid peroxidation, but not the total cytosolic ROS, was substantially upregulated by

LOXL3 depletion after stimulation of liver cancer cells with Oxaliplatin. Chemotherapeutic drugs typically kill tumors by triggering an ROS imbalance in tumor cells[28,29]. Fractionation of the cytosol and mitochondria showed that LOXL3 was partially localized in the mitochondria, and upon Oxaliplatin treatment, the mitochondrial localization of LOXL3 was enhanced. All these data should link chemotherapy to mitochondria LOXL3 mediated ferroptosis together.

We confirmed that the N-terminal peptide of LOXL3 contributes to its mitochondrial localization. The N-terminal secreted peptide sequence of the LOX family contains an amino acid sequence rich in

**Fig. 4 | Mitochondrial adenylate kinase 2 phosphorylates LOXL3 at S704 and confers resistance to chemo-induced ferroptosis. a** The workflow of the kinase screen for LOXL3-S704 phosphorylation which occurred in mitochondria. **b** The gray value was collected from the dot blot using an antibody specifically against pLOXL3-S704. The ratio of gray value by siNC versus siMK and analyzed as volcano. The red point indicates AK2. MK, mitochondrial kinase. **c** WB of Huh7 or Hep3B cells transfected with siRNAs targeting human AK2 using the indicated antibodies. **d** WB of Huh7 transfected with siRNAs targeting human AK2 and stimulated by EGF after starvation by removing serum for 12 h. **e** Huh7 or Hep3B cells expressing LOXL3-FLAG were collected for Co-IP and WB. **f** Lipid peroxidation determination of Huh7 or Hep3B cells transfected with siRNAs targeting human AK2 after treatment of Oxa-L for 12 h. **g** Huh7 cells transfected with siRNA targeting human AK2 were restored expression of gradually increasing mouse AK2 for 48 h and collected for WB. **h** LOXL3-FLAG was enriched and purified from Huh7 cells and then treated with λPP (1 μM) for half an hour to diminish S704-phosphorylation, then incubated with His-AK2 (WT or ED) in kinase reaction buffer. ED, enzyme dead, L199A/Q214A. **i, j** Huh7 or Hep3B cells transfected with siRNAs targeting human AK2 were restored expression with mouse WT or K14G/R17G mutant AK2 for 48 h and collected for WB confirmation(**i**). Meanwhile, respective lipid peroxidation determination was also performed (**j**). **k** Huh7 or Hep3B cells transfected with siRNA targeting human AK2 were restored expression of WT- or S704A-LOXL3 and treated with Oxa-L for the indicated time points and then cell viability was assessed. For (**f, j, k**), data present means ± SEM from three independent experiments (*n* = 3) and the statistical analysis was calculated by one-way ANOVA with Tukey's HSD post hoc test (**f, j**) or two-way ANOVA for multiple comparisons (**k**). Source data are provided as a Source Data file. See also Supplementary Fig. 4.

leucine that can be embedded in the phospholipid bilayer and secreted into the ECM or enter the endoplasmic reticulum[30]. Hence, we speculated that if LOXL3 regulates mitochondrial ferroptosis, it LOXL3 can enter the mitochondria and interact with mitochondrial ferroptosis-related proteins. Proteins lacking mitochondrial signal peptides can be transported into the mitochondria by mitochondrial proteins. For example, GRIM-19 functions as an integral component of complex I of the mitochondrial respiratory chain; STAT3 is imported into the mitochondria by interacting with GRIM-19[31–33]. The LOXL3 conformational structure, predicted using AlphaFold 2, revealed that amino acids 1–40 in its N-terminus LOXL3 contain a helix that bends this N-terminal peptide into a hook, exposed to the outside of the LOXL3 structure, providing the conditions for recruiting other proteins. Translocase of the outer mitochondrial membrane (TOM) is the main entry gate for nuclear-encoded mitochondrial proteins into mitochondria[34]. Under activated EGFR signaling, TOM20 functions as a guide to carry LOXL3 into mitochondria. There was a limitation in exploring the mechanism whereby the LOXL3–TOM20 axis is formed. We suspect that EGFR directly phosphorylates LOXL3 or indirectly regulates its conformational changes LOXL3 to expose its binding region to TOM20.

Exploring the upstream signaling that contributes to LOXL3 mitochondrial localization is beneficial for developing a combination strategy with Oxaliplatin to treat liver cancer. In this study, EGF-activated EGFR signaling was confirmed to drive the association of LOXL3 with the TOM complex. Blockage of EGFR signaling with the EGFR monoclonal antibody Cetuximab sensitized tumor cells to Oxaliplatin, indicating that this combination strategy should be re-evaluated in advanced liver cancer. In line with a previous report[35], the combination of Cetuximab and Lenvatinib with Oxaliplatin blocked EGFR and multiple upstream tyrosine kinase receptors, improving tumor-cell killing activity. This further verifies that inducing ferroptosis would improve tumor treatment.

There are at least three types of iron-death defense systems, based on differences in tumor-cell subcellular localization (cytoplasmic and mitochondrial GPX4, FSP1 on the cell membrane, and DHODH on the mitochondria)[20,36,37]. In this study, although GPX4 was present in the mitochondria, it was mainly localized in the cytosol, whereas DHODH was mainly localized in the mitochondria. Thus, we studied DHODH regulation, revealing DHODH-K344 as a key lysine that can be simultaneously modified by ubiquitin or oxidation. During this competitive modification process at K344, DHODH protein levels are tightly regulated. Our study further reinforces the protective role of DHODH in chemotherapy-induced mitochondrial ferroptosis.

Dimerization or multimerization is a well-documented mechanism for the activation of protein kinases and other enzymes[38–40]. LOXL3 exerts enzymatic activity by forming a homodimer[9]. Here, LOXL3-S704 phosphorylation upregulated lysyl-oxidase activity, which prevented DHODH-K344 ubiquitination. Hence, we speculated that LOXL3 phosphorylation at S704 may also regulate LOXL3 dimerization. As expected, phosphorylation at S704 significantly upregulated LOXL3

dimerization. Interestingly, LOXL3-S704 phosphorylation not only enhances lysyl-oxidase activity, but also reinforces its association with DHODH and induces its lysyl-oxidase activity on DHODH-K344. Therefore, we believe that S704 phosphorylation induces the conformational change in mitochondria LOXL3, causing it to exhibit functions that it does not have in the cytoplasm or extracellular matrix.

Comparing in the cytosol, fewer kinases reside in the mitochondria. Most mitochondrial kinases are metabolite kinases, such as adenylate kinase and those in the creatine kinase family[41]. The most well-known of these mitochondrial kinases is PTEN-induced kinase (PINK) 1[42]. To examine how LOXL3-S704 phosphorylation occurs in mitochondria, we first arranged a screening pool with 37 kinases via a literature search, using the key words "mitochondrial and kinase," including 19 metabolite kinases and 18 protein kinases that probably reside in the mitochondria or have been reported to phosphorylate mitochondrial proteins. Based on current cases of metabolite kinases with phosphorylated modified proteins, such as PKM2 and PGK1[43,44], 19 metabolite kinases were included in the screening kinase pool for LOXL3-S704 phosphorylation. This revealed that AK2 phosphorylates both metabolites as well as proteins such as LOXL3. AK2 catalyzes the interconversion of nucleoside phosphates by phosphorylating AMP and dAMP using ATP as a phosphate donor, but phosphorylates only AMP when using GTP as a phosphate donor. Comparing with other adenylate kinases, AK2 displays relatively lower broad nucleoside diphosphate kinase activity, indicating that AK2 exhibits flexibility in substrate selection[45–47]. However, it remains unknown whether AK2 utilizes proteins as substrates, although other metabolite kinases such as PKM2 and PGK1 directly phosphorylate the indicated proteins[43,44]. Our findings reveal that S704 of LOXL3 is located in the core of the beta-sheet or beta-helix, in a coil–coil state, providing probably sufficient space for AK2 to exert kinase activity on S704.

Cytosolic AK2 might not be expected to significantly increase LOXL3 phosphorylation in the cytoplasm. After restoring the expression of AK2 in AK2-depleted cells, we observed that cytosolic AK2 did not effectively upregulate LOXL3-S704 phosphorylation. Compared with mitochondrial AK2, the kinase activity of cytosolic AK2 against LOXL3 or its metabolite substrate AMP decreased significantly, suggesting that cytosolic AK2 may be deficient in the active conformation. The low kinase activity of cytosolic AK2 may be due to loss of the microenvironment that the mitochondria provide for AK2 to exhibit relatively high activity (in terms of pH and redox homeostasis), or even to posttranslational modification in the mitochondria. Another probable reason is that AK2 and LOXL3 were unable to efficiently form an axis in the cytosol, possibly owing to the loss of some scaffold proteins, especially those that exist in the mitochondria and promote the interaction of AK2 and LOXL3.

It should be pointed out that the chemotherapeutic agents such as oxaliplatin or 5-Fu from FOLFOX in this study, is not a global strategy for HCC but rather used in selected patients. Chemosensitization can effectively curb tumor-cell escape and avoid serious side-effects. For this purpose, we used low-dose Oxaliplatin with Leflunomide, a

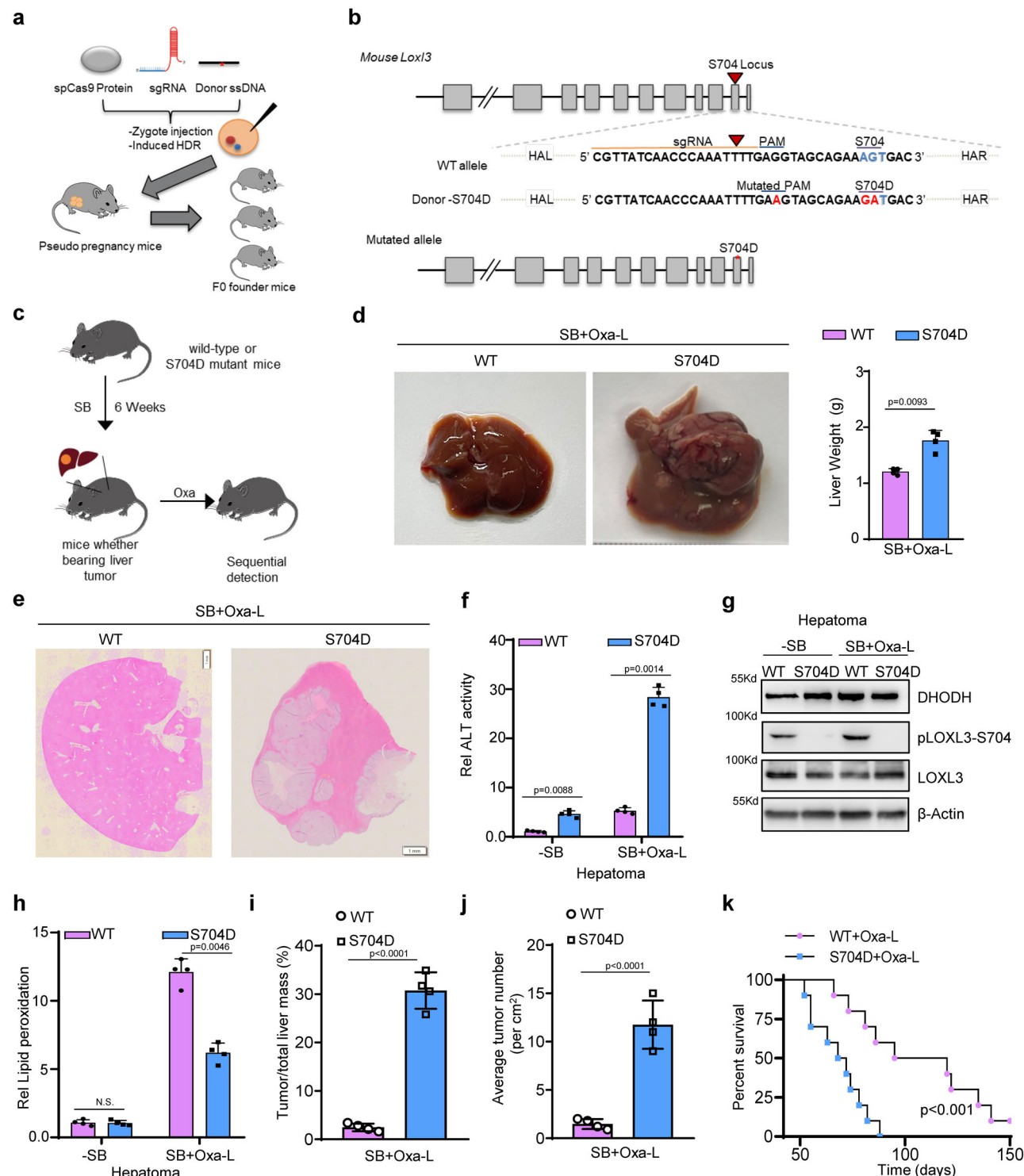

**Fig. 5 | The Loxl3-S704D mutant confers resistance to Oxaliplatin treatment in sleeping beauty transposon-induced liver cancer. a, b** For *Loxl3-S704D* mutant mouse generation, CRISPR–Cas9 mRNA, sgRNA and mutation donor ssDNA mixture were injected into the cytoplasm of fertilized eggs transferred into oviducts of pseudo pregnant female mice at 0.5 dpc for the birth of mutated mice. **c** 6-week-old male WT or *Loxl3-S704D* mutant mice were injected with sleeping beauty (SB) transposons carrying oncogenic dNβ-catenin and c-Met in the tail. The mice received low-dose Oxaliplatin (1 mg kg⁻¹ bodyweight) treatment intraperitoneally 3 times a week for 14 days. *n* = 4 in each group. **d**–**j** After 6 weeks induction and the following treatment, the representative livers of WT and *Loxl3-S704D* mutant mice are displayed, and liver weight was measured (**d**). HE staining was performed to

ascertain tumor occurrence (**e**). The livers were further used for determinations of ALT activity (**f**), WB using indicated antibodies (**g**) and the lipid peroxidation levels (**h**). The ratio of tumor to liver mass (**i**) and the average tumor number (**j**) were analyzed and calculated. *n* = 4 in each group. **k** Another batch of mice receiving intraperitoneal low-dose Oxaliplatin (1 mg kg⁻¹ bodyweight) treatment was used to calculate the survival time by Kaplan–Meier plotter. *n* = 10 in each group. For (**d**, **f**, **h**–**k**), data represent means ± SEM of individual mouse groups (*n* = 4 or 10) and the statistical analysis was calculated by two tailed Student's *t* test (**d**, **i**, **j**), one-way ANOVA with Tukey's HSD post hoc test (**f**, **h**) or log-rank test (**k**). Source data are provided as a Source Data file. See also Supplementary Fig. 5.

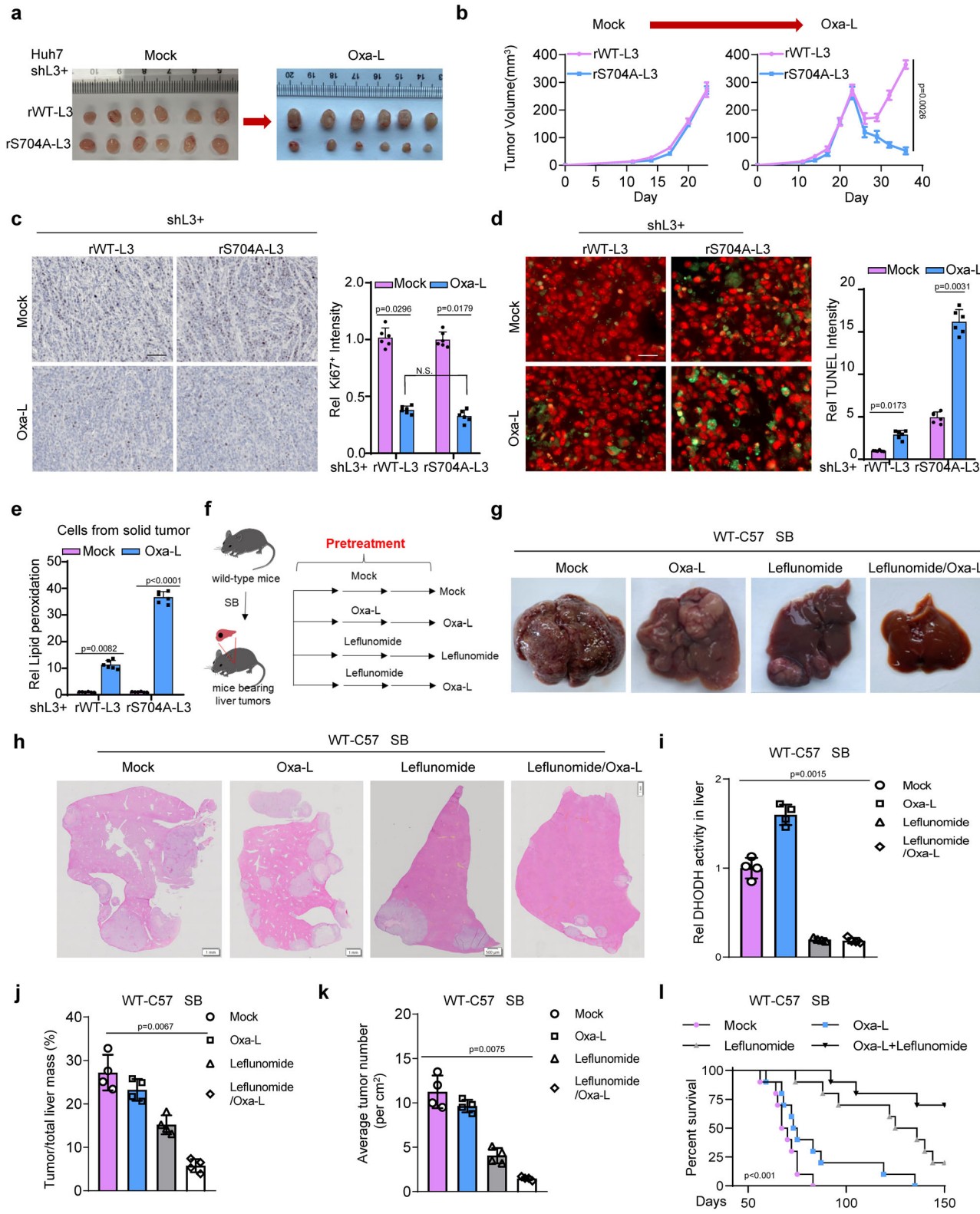

DHODH inhibitor, to block the AK2–LOXL3–DHODH axis in liver cancer. A better combination strategy might be low-dose Oxaliplatin plus a LOXL3 inhibitor; however, as mentioned, inhibitors targeting LOXL3 have not been effectively developed. We therefore chose low-dose Oxaliplatin, along with Leflunomide, which is used to treat rheumatoid arthritis. In both *S704D-Loxl3* knock-in mice and transplanted tumor models with S704A-LOXL3, LOXL3 was found to be an important counterpart to low-dose Oxaliplatin for the treatment of liver cancer.

In conclusion, these findings reveal that LOXL3, a member of the LOX family, participates in anti-mitochondrial ferroptosis via its high mitochondrial activity levels which are upregulated by AK2. The high activity of LOXL3 competes to occupy DHODH-K344 and attenuates DHODH ubiquitin at K344, in turn inducing the accumulation of mitochondrial DHODH. This further confers resistance to mitochondrial ferroptosis induced by chemotherapeutic drugs such as Oxaliplatin. Finally, low-dose Oxaliplatin combined with Leflunomide, a

**Fig. 6 | The combination strategy of Oxaliplatin with Leflunomide efficiently facilitates the treatment of advanced liver cancer. a–e** Xenograft study. Xenogeneic inoculation of Huh7 LOXL3-depleted cells restored with WT- or S704A-LOXL3 was performed into the left groins of nude mice (*n* = 6 in each group). When the tumor size reached 50 mm² (approximately 3 weeks after inoculation), low-dose Oxaliplatin (1 mg kg⁻¹ bodyweight) was administered intraperitoneally 3 times a week for 14 days. The solid tumors formed by Huh7 cells with the indicated treatment or control are shown (**a**). The tumor growth rate (**b**) was recorded and analyzed. The tumors were collected and fixed and then stained with an antibody against Ki67 (**c**) or TUNEL (**d**) or lipid peroxidation determination and calculation (**e**). Scale bars: 1 mm (**c**), 200 μm (**d**). **f** Experimental design for SB-driven hepatocarcinogenesis for treatment validation. SB induced liver tumors were preliminarily achieved within 8 weeks for wild-type C57 mice. The DHODH inhibitor Leflunomide (0.2 mg kg⁻¹ bodyweight) or low-dose Oxaliplatin (1 mg kg⁻¹ bodyweight) was

intraperitoneally pre-administered for 12 days with once every 4 days as the experimental group, while the mock group was used as the control. After the pre-administration, Oxaliplatin (1 mg kg⁻¹ bodyweight) or DHODH inhibitor Leflunomide (0.2 mg kg⁻¹ bodyweight) was intraperitoneally treated for 12 weeks with 3 times a week. **g–k** livers were dissected (**g**). HE staining was performed using these representative livers (**h**). DHODH activity in liver tumor was tested (**i**). The ratio of tumor to liver mass (**j**) and the average tumor number (**k**) were analyzed and calculated. n = 4 in each group. **l** Another batch of mice receiving the indicated treatments or control mice was used to calculate the survival time by Kaplan–Meier plotter. *n* = 10 in each group. For (**b–e**, **i–l**), data represent means ± SEM of individual mouse groups (*n* = 4, 6 or 10) and the statistical analysis was calculated by two-way ANOVA for multiple comparisons (**b**), one-way ANOVA with Tukey's HSD post hoc test (**c, d, i–k**) or log-rank test (**l**). Source data are provided as a Source Data file. See also Supplementary Fig. 6.

DHODH inhibitor, efficiently compromised liver cancer progression, prolonging survival time in mice.

## Methods

### Clinical specimens
The use of pathological specimens, as well as the review of all pertinent patient records, was approved by the Research Ethics Committee of Zhuhai People's Hospital. A total of 60 liver cancer samples from HCC (primary human hepatocellular carcinoma) patients, including 49 male and 11 female, were collected during surgical treatment, following with the FOLFOX (Oxaliplatin plus fluorouracil and leucovorin) chemotherapy regimen in Zhuhai People's Hospital, between Jan 2010 to Dec 2015. All clinical samples were collected with informed consent under Health Insurance Portability and Accountability Act (HIPAA)-approved protocols. The summary clinical information about the patients of the cohort is supplied in the Supplementary Table 1. Additionally, the clinical information about patients whose tumor tissues were used for PDX transplantation is supplied in Supplementary Table 2. The authors affirm that human research participants provided informed consent to participate in the study and for publication of their data in Supplementary Table 2.

### Cellular ROS and lipid peroxidation level determination
Cellular ROS levels were measured using an ROS detection kit (BioVision, Waltham, MA; #K936-100), as previously described[48]: 50,000 cells were plated in wells and collected, and fluorescence intensity at Ex/Em = 495/529 nm was quantified using a BioTek Synergy Neo Multi-Mode Plate Reader (BioTek, Santa Clara, CA). A representative plot of three biological replicates is shown.

For lipid Peroxidation determination, cells were seeded in DMEM and washed twice in HBSS and treated with Oxaliplatin (Normal-dose, 5 μg ml⁻¹; low-dose, 1 μg ml⁻¹) in DMEM for the indicated time, then incubated in DMEM containing 2 mM BODIPY 581/591 C11 (Invitrogen, D3861) for 30 min at 37 °C. For imaging, slides were excited using the 488 and 565 nm laser and fluorescence measured from 505 to 550 nm and above 580 nm using Zeiss LSM880 microscope. Upon oxidation of the polyunsaturated butadienyl portion of the dye, there is a shift of the fluorescent emission peak from 590 nm to 510 nm, remaining lipophilic and thus reflecting lipid peroxidation. Further examination was performed through a BD laser analyzer using PE-Texas Red (PE-TR) filter (measuring non-oxidized BODIPY-C11) and fluorescein isothiocyanate (FITC) (measuring oxidized BODIPY-C11). A minimum of 10,000 cells were collected using BD FACS Diva v8.0 in each condition and analyzed using FlowJo v10.7 (Bioscience).

### Subcellular fractionation
To obtain the subcellular fractions, Huh7 or Hep3B cells were harvested and washed three times with cold PBS. Cytosolic and nuclear fractions were obtained first, depending on the osmotic

pressure tolerance. Mitochondrial fractions were prepared using the Mitochondria/Cytosol Fractionation Kit (BioVision, K256).

### Immunoprecipitation (IP) and LC-MS/MS of posttranslational modification
Immunoprecipitation (IP) for FLAG-tagged LOXL3 from Huh7 or Hep3B cells, and separation via SDS–PAGE gel, were as previously described[49]. Samples from SDS-PAGE gels were digested with trypsin, and peptides were analyzed using a nanoflow HPLC Easy-nLC 1000 system coupled to a Q Exactive HF mass spectrometer (Thermo Fisher Scientific). The MS raw data were analyzed using Proteome Discoverer 2.3 against the human Swiss-Prot database containing 20,231 sequences (downloaded in December 2017). Phospho-peptide matches were analyzed using MaxQuant 1.5.2.8, implemented in Proteome Discoverer, and were manually curated[50].

### LOXL3 enzymatic assay
To analyze the activity regulation of LOXL3 and of the DHODH-K344 peptide ALEK₃₄₄IRAGAS, and an oxidation assay was performed as previously described[4,51]. Briefly, we purified His-tagged LOXL3 from *Escherichia coli* or FLAG-tagged LOXL3 from mammalian cells, then concentrated the LOXL3 protein to 200 μg ml⁻¹. Activity assays were performed using peptides at 10 mg ml⁻¹. The enzymatic reaction was initiated by adding a substrate mixture containing 50 mM sodium borate (pH 8.2), 100 mM N-acetyl-3,7-dihydroxyphenoxazine, and 20 mM 1,5-diaminopentane. The production of hydrogen peroxide by LOXL3 results in fluorescent resorufin production, which can be measured by excitation at 540 nm and emission at 580 nm. The fluorescence reaction was measured every 5 min for 1.5 h at 37 °C using a BioTek Synergy Neo Multi-Mode Plate Reader (BioTek). Each representative bar data point was derived from three independent replicates.

### In vitro kinase assay
LOXL3-FLAG was purified from Hep3B cells stably expressing WT LOXL3-FLAG, washed with cold PBS containing 2% NP40, and treated with λPP (1 μg ml⁻¹) for half an hour. His-AK2 was purified from *Escherichia coli* cells, and was concentrated to 1 mg ml⁻¹. Next, 200 ng of His-AK2 and 1 μg of FLAG-LOXL3 were incubated in 25 μL of kinase reaction buffer (CST, Danvers, MA, USA, #9802) for 30 min. The reaction was terminated by adding sample buffer and boiling for 4 min at 95 °C in SDS-loading buffer. Western blotting was performed to measure the level of LOXL3 phosphorylation at S704, using the indicated antibodies.

### Experimental mice
All animal studies were conducted on mice at about six weeks of age. Mice were group-housed (4-5 mice per cage). All mice were maintained under a 12-hour light/dark cycle with free access to water and standard mouse chow in a specific-pathogen-free (SPF) facility. All

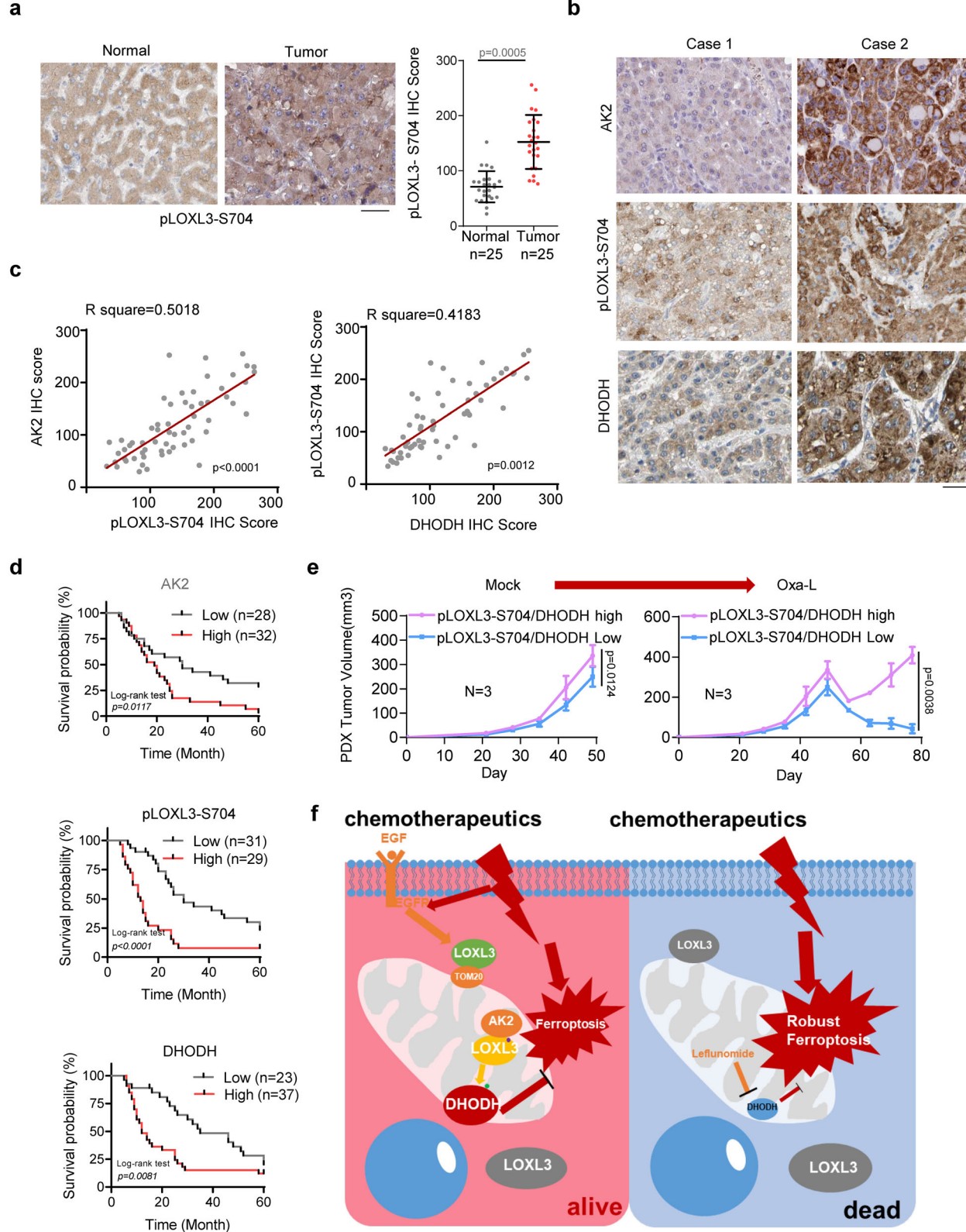

mice received humane care, and all experimental procedures were performed in compliance with the Guide for the Care and Use of Laboratory Animals and approved by the Institutional Animal Care and Use Committee (IACUC) in Guangzhou University.

For the xenograft study, $2 \times 10^6$ Huh7-derived stable cells were subcutaneously injected into the left groins of randomized 6-week-old female athymic nude mice. Approximately three weeks after

inoculation when the tumor size reached 50 mm[2], low dosage of Oxaliplatin (1 mg kg[−1] bodyweight) administered intraperitoneally 3 times a week for 14 days. After the treatment, mice were euthanized, and the tumors were dissected. Differences in the growth rate, tumor volume, cell proliferation, apoptotic percentage or lipid peroxidation levels in xenograft tumors were statistically analyzed. Twice a week, the tumor size was measured and recorded using calipers in two

**Fig. 7 | The AK2/LOXL3/DHODH axis predicts the prognosis of liver cancer patients and supplies a combination strategy for advanced liver cancer. a** IHCs of human HCC clinical samples ($n = 50$) were performed using antibodies against pLOXL3-S704 or LOXL3. Representative images are shown. The pLOXL3-S704 IHC score was obtained for analysis and calculation. Scale bars: 200 μm. **b**, **c** IHCs of human HCC clinical samples were performed using antibodies against pLOXL3-S704, and DHODH IHC is shown in Fig. S7b-c, respectively. Representative images are shown in (**b**). The correlated IHC signal between AK2 and pLOXL3-S704, pLOXL3-S704 and DHODH in human HCC samples was calculated (**c**). Scale bars: 200 μm. **d** Kaplan–Meier plot of survival based on AK2, pLOXL3-S704 and DHODH expression in human HCC. IHCs of human HCC clinical samples are shown in (b–d). The overall survival of human HCC patients according to AK2, pLOXL3-S704 and DHODH IHC scores was analyzed by Kaplan–Meier plots. **e** Low-dose Oxaliplatin (1 mg kg$^{-1}$ bodyweight) treatment test using Patient-Derived Xenograft (PDX)

Models. Human HCC samples were collected for PDX preparation, according to the histological expression: high expression of pLOXL3-S704 and DHODH ($n = 3$ patients), or low ($n = 3$). Tissues were prepared in a volume of 5–10 mm$^3$, and then subcutaneously implanted into the left flanks of 6-week-old female NSG mice following standard procedures. Mice were treated with low-dose Oxaliplatin via intraperitoneal injection every 3 times a week for 1 month. The tumor size was measured as previously described. **f** Schematic model of AK2/LOXL3/DHODH axis-mediated resistance to chemotherapy drug-induced ferroptosis. For (**a**, **e**), data represent means ± SEM of individual human HCC sample groups (**a**) or the PDX groups (**e**). The statistical analysis was used by two-tailed Mann–Whitney $U$ test (**a**), two-tailed Student's $t$ test for Pearson correlation coefficient (**c**), the log rank test (**d**) or two-way ANOVA for multiple comparisons (**e**). Source data are provided as a Source Data file. See also Supplementary Fig. 7.

dimensions to generate a tumor volume using the following formula: $0.5\times$ (length $\times$ width$^2$).

For PDX transplantation, frozen PDX tumors collected from the tumors of human HCC patients were prepared in a volume of 5-10 mm$^3$, and then subcutaneously implanted into the flanks of 6-week-old NSG female mice following standard procedures. Mice were treated with low-dose Oxaliplatin (1 mg kg$^{-1}$ bodyweight) via intraperitoneal injection every 3 times a week for 1 month. The tumor size was measured weekly using calipers in two dimensions to generate a tumor volume using the following formula: $0.5 \times$ (length $\times$ width$^2$).

For *Loxl3-S704D* mutant mouse generation, Cas9 mRNA and sgRNA were generated by in vitro transcription and mutation donor ssDNA mixture were injected into the cytoplasm of fertilized eggs transferred into oviducts of pseudo-pregnant female mice at 0.5 dpc for the birth of mutated mice. The *Loxl3-S704D* mouse line was generated on a mixed FVB/N and C57BL/6 background. C57BL/6 mice were used for MET/CAT model construction in the proteomic study.

For hydrodynamic injection, plasmids (Met: PT3EF1aH-hMet; Ctnnb1/b-catenin: PT3EF1aH-b-catenin; sleeping beauty transposase (SB): pCMV/SB) were kindly provided by Professor Lijian Hui. Oncogene-expressing constructs were delivered via hydrodynamic tail-vein injection into male mice at six weeks of age, as previously described[52–56]. The design of in vivo administration of Oxaliplatin (low-dose: 1 mg kg$^{-1}$ bodyweight) alone or in combination to treat SB-driven hepatocarcinogenesis is described as follows: SB induced liver tumors were preliminarily achieved within 6 weeks in Fig. 5c, 8 weeks in Fig. 6g and 8 weeks in Supplementary Fig. S6b. The DHODH inhibitor Leflunomide (0.2 mg kg$^{-1}$ bodyweight) or low-dose Oxaliplatin (1 mg kg$^{-1}$ bodyweight) was intraperitoneally pre-administered for 12 days with once every 4 days as the experimental group, while the mock group was used as the control. After the pre-administration, Oxaliplatin (1 mg kg$^{-1}$ bodyweight) or DHODH inhibitor Leflunomide (0.2 mg kg$^{-1}$ bodyweight) was intraperitoneally treated for 12 weeks with 3 times a week. After the treatment, mice were euthanized, and the tumors were dissected. Tumor volume and weight, ratio of tumor to liver mass, the average tumor number or lipid peroxidation levels in HCC tumors were recorded, measured and statistically analyzed.

**Statistical analysis**

All experiments were repeated independently with similar results from three times and presented as mean ± standard error of the mean (SEM) unless otherwise specified. Statistical analyses were performed using two tailed Student's $t$ test for pairwise comparison, one-way or two-way analysis of variance (ANOVA) with Tukey's HSD post hoc test for multiple comparisons and Mann–Whitney $U$ test for comparison of clinical patients on GraphPad Prism 9. For the in vivo and ex vivo experiments, mice were randomly assigned to different groups. Other experiments were not randomized. No statistical method was used to pre-determine the sample size. No data were excluded from the analyses. The investigators were not blinded to allocation experiments

and outcome assessment except the histology quantification. For genetically engineered mouse modes (GEMMs) analysis and the xenograft assays, the examinations were results of one-time experiment with sufficient animal number for statistical analysis indicated in figure legend. n values indicated biologically independent experiments or samples. $p < 0.05$ was considered statistically significant. In all figures, not significant (N.S.), $p < 0.05$ ($*$), $p < 0.01$ ($**$), $p < 0.01$ ($***$).

## Reporting summary
Further information on research design is available in the Nature Portfolio Reporting Summary linked to this article.

## Data availability
All data, including original WB and statistical data, were released in the Source Data file. Source data are provided with this paper.

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

## Acknowledgements

This study was mainly supported by Guangdong Natural Science Foundation of Guangdong Province of China 2021B1515020016 and

2022B1515020010, in part by NSFC grants No. 92053113, 82230067, 81902863, 82103511, 81901857, 82102961.

## Author contributions

X.W. and L.L. designed the experiments while Y.H. prepared the manuscript. M.Z., H.Y., and Y.D. performed most experiments and analyses. Y.L. and J.X. performed a specific subset of the experiments and contributed to the computational statistical analysis. S.H., M.Z., and Y.D. performed the pathology analyses.

## Competing interests

The authors declare no competing interests.
