## [Peer Review File · Nature Communications]

Lysyl oxidase-like 3 restrains mitochondrial ferroptosis to promote liver cancer chemoresistance by stabilizing dihydroorotate dehydrogenaseREVIEWER COMMENTS

Reviewer #1 (Remarks to the Author); expert in signaling and ferroptosis:

In this study, the authors proposed a model wherein EGF stimulation promotes LOXL3 localization into mitochondria, where AK2 phosphorylates LOXL3 at S704, and this phosphorylation promotes LOXL3's enzymatic activity to subsequently catalyze K344 oxidation of DHODH, which prevents DHODH from K344 ubiquitination and degradation; DHODH protein stabilization suppresses chemotherapy drug Oxaliplatin-mediated ferroptosis. They further showed that S704D LOXL3 mice confers Oxaliplatin resistance in liver cancer models, whereas LOXL3 deficiency or DHODH inhibitor treatment sensitizes liver cancers to oxaliplatin treatment.

Overall, this is an interesting study with a lot of data ranging from detailed mechanistic studies to KI animal studies and patient sample analyses. The findings are novel and provide important insights for both mechanistic understanding ferroptosis and disease treatment. However, because the proposed model is quite complicated, the study also exposes quite some weaknesses which need to be addressed.

1. At multiple places in this manuscript, the authors compared phenotypes between LOXL3 knockdown cells with LOXL3 WT and mutant restoration. In these analyses, they need to include control and knockdown cells, including (but not limited to):

Fig. 1F: here the authors need to compare protein levels in four sets of cells, WT, LOXL3 KO, LOXL3 KO with restoration of LOXL3 WT or KD (as used in Fig. 1C).

Likewise, in Fig. 2H-J, the authors need to add WT and LOXL3 KO cells.

Fig. 3E: the authors need to also compare CHX profiles of DHODH between WT and LOXL3 knockdown cells.

"As Figure 3B and 3C showed, DHODH protein level but not its mRNA was downregulated in LOXL3 knockdown cells" The corresponding data in Fig. 3B-C did not compare WT and LOXL3 knockdown cells. They need to perform the experiments as described in the text (by adding WT and knockdown cells).

2. Fig. 1J, K, L: "we constructed cell lines stably expressing wild-type or enzymatically dead (ED) forms of LOXL3 with mitochondrial signal peptides (named M-WT-L3 or M-ED-L3) in LOXL3-deficient cells (Figure S1G-H)." Since LOXL3 can localize in mitochondria, what is the point to add a mitochondrial signal peptides to LOXL3? It seems that their data in Fig. 1J, K did not prove anything, since the effect of expressing WT-L3 and M-WT-L3 is similar. Also, they did not include ED-L3 here, and there is no comparison for M-ED-L3.

The authors need to compare mitochondrial localization (by fractionation as shown in Fig. 1H) of WT-L3 and M-WT-L3 (as well as ED-L3 and M-ED-L3). I assume LOXL3 does not have the classic mitochondrial localization signal, so adding a mitochondrial localization signal to LOXL3 should increase its localization into mitochondria. If so, why there is no functional difference between WT-L3 and M-WT-L3? What does the story gain by showing this data?

3. Protein levels of LOXL3 and DHODH in cell lines and experimental conditions in Fig. 3J need to be shown. (There is a misspelling in Fig. 3J "DHOD-Myc".)

4. Is ferroptosis at least partly responsible for the decreased cell viability and increased cell death in LOXL3 knock-down cells with Oxa treatment (Fig. 1 and Fig. S1C)? This can be addressed by examining whether adding the ferroptosis inhibitor ferrostatin-1 or liproxstatin-1 can rescue cell death and restore cell viability in this context.

5. Fig. 1, Fig. 4F: the authors need to confirm their major findings with at least two si/shRNAs for LOXL3 and AK2.

6. Fig. 5G: under Oxa treatment condition, the levels of DHODH in WT and S704D samples are the same. If so, how to explain the difference in lipid peroxidation and tumor phenotypes between WT and S704D mice under Oxa treatment (Fig. 5H-K)?

Minor comments:

The manuscript writing needs to be significantly improved. Suggest the authors to seek help from a professional writing editor.

Also, in the Introduction, they mixed background introduction with their findings, which is confusing (in Introduction of most other papers, the authors first introduce relevant background and highlight unmet needs and knowledge gaps, and briefly summarize their findings in the last paragraph of Introduction). Introduction needs to be re-drafted.

It is really painful to examine their clonogenic survival data, which contain multiple lines with similar colors. The authors can add other features (such as solid and dash lines) to distinguish different groups.

Reviewer #2 (Remarks to the Author); expert in liver cancer and chemoresistance:

The proposed manuscript of Wang et al. investigated the molecular mechanisms of drug resistance in hepatocellular carcinoma (HCC) driven by Lysyl oxidase-like 3 protein (LOXL3) and its regulatory elements. The authors propose that resistance to Oxaliplatin is initiated by EGFR signaling and propagated by AK2-LOXL3-DHODH axis in mitochondria which ultimately affects the rate of ferroptosis. They identified that LOXL3 exerted high activity in mitochondria and is regulated by AK2 kinase. Once activated, LOXL3 prevents ubiquitination of DHODH which consequently decreases mitochondrial ferroptosis and displays higher resistance to chemotherapy. Drug resistance represents an increasing obstacle for patients with HCC, therefore this study focuses on a timely and relevant topic. The project is technically well executed and provides novel mechanistic insights into drug resistance in HCC. Utilization of the cell lines, animal models, and human samples provided potential translational value. However, there are several questions and open issues that limit the enthusiasm for the manuscript. Further, several issues related to the selection of the drug, exploration of underlying mechanism as well as lack of independent external validation exist.

Major comments:

- The selection of platin-based compounds for the evaluation of liver cancer drug resistance is unclear. It should be clearly explained and demonstrated if LOXL3 exerts general properties of chemoresistance or if a platin-dependent effect is proposed. Herein, given that chemotherapy plays a minor role in HCC treatment it would be beneficial to validate results with approved drugs for HCC in the context of LOXL3. In line with this, Sorafenib is reported to induce autophagy, apoptosis, and activates ferroptosis, which might make this compound particularly promising. Overall, the rationalization of the Oxaliplatin investigation as well as the focus on liver cancer should be delineated in more detail and the rationale should be provided for the actual human diseases.

The authors mention that oxaliplatin causes massive cell death in shLOXL3 cells which prevents detailed investigation of ferroptosis. Given that viability is generally over 50% (Figure 1), this is difficult to understand and should be explained.

- The mechanism of how EGF/EGFR activation regulates LOXL3 remains uncertain and should be experimentally addressed. How does EGFR activate TOM20 protein and initiate translocation of LOXL3 inside mitochondria?

o If EGFR-driven chemoresistance to Oxaliplatin is the key molecular mechanism, can this be validated by inhibition of EGF/EGFR pathway? Can results of LOXL3 be recapitulated by combining EGFR inhibitor and Oxaliplatin, i.e. increased level of ferroptosis and increased cell death. Moreover, tyrosine kinase inhibitors that are approved for HCC (e.g. Sorafenib, Lenvatinib), affect

EGFR signaling which should be explored in the context of the study.

o To confirm the role of ferroptosis in this process, it would be interesting to assess direct activation of ferroptosis rather than solely using indirect activation by chemotherapeutic compound.

- The relevance of the suggested findings for human HCC is interesting. The investigated cohort should be described in more detail and clinico-pathological information for the described patients (Figure 7) should be provided. Can the authors confirm an association to response to the therapy, e.g. histologically, depending on the expression of AK/LOXL3/DHODH?

- The suggested prognostic implication is interesting. Independent validation in the external dataset would be interesting. In addition, predictive impact for therapy other than oxaliplatin should be demonstrated.

- The validation of the findings in PDx is interesting and relevant. The clinic-pathological information should be provided and the investigations should be described in more details. In particular, it is unclear how many tumor specimens were processed and how the two groups were established.

- Consistency in data representation is unclear (the result section mainly presents data mostly for Hep3B cell line). The results should be consistently presented throughout for at least both of the cell lines.

Minor comments

- The provided abstract requires streamlining and simplification for readers not familiar with the topic

- Results from the animal model in Figure S6A - the effects between Oxaliplatin and LOXL3 inhibitors look additive rather than synergistic. Following the claims of the authors, synergism is expected. Please elute.

- Figure 6I shows no difference between the Leflunomide and combined Leflunomide/Oxaliplatin. It would be expected that DHODH activity would be further decreased when used in combination. Please comment.

- What is the reason to use female mice for xenograft studies and male mice for HTVI-induced HCCs

Reviewer #3 (Remarks to the Author); expert in lysyl oxidases:

In this manuscript, Zhan et al report that lysyl oxidase-like 3 (LOXL3), but not other members of LOX family, confers resistance to oxaliplatin in liver cancer cell lines. They showed that upon depletion of LOXL3, liver cancer cell lines become responsive to oxaliplatin by increasing lipid peroxidation and inducing ferroptosis. The role of LOXL3 in oxaliplatin resistance in vitro has been validated using a robust in vitro add-back system where authors stably knocked down LOXL3 and overexpressed different versions of LOXL3, including wild type, enzymatically dead, mitochondria-directed ones, etc. They showed that EGF induces the localization of LOXL3 in mitochondria by inducing its interaction with TOM20, a member of mitochondrial outer membrane. The phosphorylation and activation of mitochondrial LOXL3 was shown to be mediated by mitochondrial adenylate kinase 2 (AK2) at S704 site, and this phosphorylation was shown to confer resistance to chemotherapy-induced ferroptosis. The authors also demonstrated that LOXL3 phosphorylation and activation prevent ubiquitination of DHODH (a key mitochondrial protein involved in several metabolic process) by oxidizing the lysine residues on DHODH leading its stability, which ultimately results in resistance to oxaliplatin-induced ferroptosis. Although not clear, authors also showed S704D mutation (inducing LOXL3 homodimerization) on LOXL3 confers chemoresistance in a liver tumor model. In addition, using xenografts and PDXs, they examined the contribution of S704A

mutation on oxaliplatin resistance in vivo. Finally, they tested DHODH inhibitor together with oxaliplatin in a liver tumor model and examined the expression of AK2/pLOXL3/DHODH axis in liver patients' tissues by immunohistochemistry.

While the manuscript reports an interesting non-canonical function of LOXL3 where mitochondrial LOXL3 and its mitochondrial-specific phosphorylation plays role in ferroptosis and potentially in chemoresistance, there are major concerns which need to be addressed.

1. It is not clear why authors have chosen two liver cancer cell lines (Huh7 and Hep3B) for this study. Are they resistant to oxaliplatin, which is the major chemotherapy used in this study? Furthermore, it is not clear why oxaliplatin, but not other chemotherapy agents used in liver cancer, is chosen throughout the study. If the authors claim that LOXL3 is critical for "chemoresistance" in general, they should repeat some of the experiments with other chemotherapy agents.

2. While the authors claim that chemotherapy-induced EGFR signaling is responsible for the mitochondrial LOXL3 expression, it is not shown in this paper. This is critical as the upstream of the LOXL3 activation is shown to be via EGF signaling. In its current form, there is a major disconnect from oxaliplatin to EGFR activation to mitochondrial LOXL3 induction. Furthermore, it is not clear if oxaliplatin changes mitochondrial LOXL3 activity (compare Supp Fig 1J and L). This needs to be clarified.

3. Another disconnect is the identification of AK2 as the mitochondrial kinase phosphorylating LOXL3 on S704 site. Although AK2 phosphorylation of LOXL3 is convincing, it is not clear if LOXL3 phosphorylation by AK2 is EGF dependent. In other words, is its phosphorylation by AK2 mediated by chemotherapy-induced EGFR activation? If not, it is hard to claim the axis presented here as responsible mechanism for mitochondrial LOXL3 mediated chemoresistance.

4. DHODH is a key protein playing roles in both pyrimidine biosynthesis and mitochondrial respiratory chain. Although the study assumes DHODH as proxy for the lipid peroxidation and ferroptosis, it is not clear what DHODH stabilization upon LOXL3 activation does on these other key metabolic processes which are also potential major mechanisms involved in chemoresistance. This needs to be addressed experimentally. Also, while the authors measured the cytoplasmic ROS and did not observe a major change upon LOXL3 modulation, what happens to mitochondrial ROS which is also controlled by DHODH in mitochondria?

5. Overall, the in vivo experimental set-ups are not clear at all, and the results section describing those findings are not written well. Indeed, the in vivo experiments are missing the validation of key in vitro findings. Importantly, testing DHODH inhibitor together with oxaliplatin is again missing the context of the manuscript with respect to LOXL3 involvement. A better experimental design is needed to show that DHODH is a key contributor to LOXL3-driven chemoresistance in vivo. Furthermore, treatment schemes, treatment duration, doses, and sample sizes are confusing across the board.

6. It is surprising that the authors generated CRISPR knock-in mice of LOXL3 S704D instead S704A. What is the rationale for this? The reviewer appreciates the use of the system and the model, but it is hard to grasp why this mutation is chosen. Indeed, the author's rationale provided in the beginning of the Results section describing this experiment is vague. What is the physiological or pathological relevance of this specific mutation? Furthermore, key comparisons (groups) are missing in this experiment. For example, what is the impact of S704D on tumor growth without chemotherapy?

7. It is not clear how much of chemosensitization is attributed to the mitochondrial function of LOXL3 in a systemic treatment setting. Is it totally independent of its canonical collagen crosslinking function in the tumor microenvironment? Authors mentioned PXS-5153A (a clinically tested LOXL2/3 inhibitor) in the M&M section; however, there is no data shown with this inhibitor. In vivo testing of LOXL2/3 inhibitor in one of the xenograft models in combination with oxaliplatin and downstream analysis of lipid peroxidation and collagen cross-linking/drug penetration/signaling impact is needed.

8. The results section can be substantially shortened by removing unnecessary explanations, Furthermore, the rationale of the experiments should be better defined in each section. The discussion section is written like a more "justification" section than the discussion section without citing key studies related to topic of this study. Furthermore, the limitations of the study should be provided.

Minor points:

1. Figure 1I is not mentioned in the text.
2. Western blot images need to have the molecular weight of the proteins shown next to the images.
3. It is not clear if 5-FU or oxaliplatin is used in Figure S1L.
4. The impact of stable knockdown of LOXL3 on the other LOX family members need to be shown at mRNA and protein levels.
5. Statistical tests and comparison groups needs to be clearly stated.
6. The details of the PDX models used need be provided.

RESPONSE TO REVIEWERS' COMMENTS

According to comments and suggestions, we improved this study. All changes made in the manuscript are marked as yellow.

Reviewer comments:

Reviewer #1:

In this study, the authors proposed a model wherein EGF stimulation promotes LOXL3 localization into mitochondria, where AK2 phosphorylates LOXL3 at S704, and this phosphorylation promotes LOXL3's enzymatic activity to subsequently catalyze K344 oxidation of DHODH, which prevents DHODH from K344 ubiquitination and degradation; DHODH protein stabilization suppresses chemotherapy drug Oxaliplatin-mediated ferroptosis. They further showed that S704D LOXL3 mice confers Oxaliplatin resistance in liver cancer models, whereas LOXL3 deficiency or DHODH inhibitor treatment sensitizes liver cancers to Oxaliplatin treatment.

Overall, this is an interesting study with a lot of data ranging from detailed mechanistic studies to KI animal studies and patient sample analyses. The findings are novel and provide important insights for both mechanistic understanding ferroptosis and disease treatment. However, because the proposed model is quite complicated, the study also exposes quite some weaknesses which need to be addressed.

Response: We thank a lot for the reviewer's positive comments, insightful criticism and constructive suggestions which heavily strengthen our study. As such, we performed requested experiments by the reviewer and revised the manuscript to improve the clarity.

Main comments:

1. At multiple places in this manuscript, the authors compared phenotypes between LOXL3 knockdown cells with LOXL3 WT and mutant restoration. In these analyses, they need to include control and knockdown cells, including (but not limited to):

1) Fig. 1F: here the authors need to compare protein levels in four sets of cells, WT, LOXL3 KO, LOXL3 KO with restoration of LOXL3 WT or KD (as used in Fig. 1C).

Response: Thanks for the reviewer's mention. According to his or her suggestion, in **Fig. 1f**, we compared protein levels in four indicated sets of cells the reviewer requested.

2) Likewise, in Fig. 2H-J, the authors need to add WT and LOXL3 KO cells.

Response: Thanks for the reviewer's mention. According to his or her suggestion, in **Fig. 2h-j**, we added the indicated sets of cells the reviewer requested.

3) Fig. 3E: the authors need to also compare CHX profiles of DHODH between WT and LOXL3 knockdown cells.

Response: Thanks for the reviewer's mention. According to his or her suggestion, we supplemented data of comparing CHX profiles of DHODH protein level between WT and LOXL3 knockdown cells in **Supplemental Fig. S3e**.

4) *"As Fig. 3B and 3C showed, DHODH protein level but not its mRNA was downregulated in LOXL3 knockdown cells" The corresponding data in Fig. 3B-C did not compare WT and LOXL3 knockdown cells. They need to perform the experiments as described in the text (by adding WT and knockdown cells).*

Response: Thanks for the reviewer's mention. According to his or her suggestion, in **Fig. 3b-c**, we added the compare result of WT and LOXL3 knockdown cells to correspond what was described in the text.

Summary Response to this main comment: We apologize for not adding enough control groups when firstly compared phenotypes between LOXL3 knockdown cells with LOXL3 WT and mutant restoration, making it not clear and solid enough in the initial manuscript. Now, in the revised manuscript, we added WT and LOXL3 knockdown cells when compared phenotypes between LOXL3 knockdown cells with LOXL3 WT and mutant restoration, especially at the first time to compare, like in **Fig. 1(e-f, j-k)**, **Supplemental Fig. S1(i-j, o-p, s)**, **Fig. 4i-j**, which made the conclusion to be more reliable. It should be pointed out that, based on the supplemented reliable phenotypes of WT and LOXL3 knockdown cells in the revision and our understanding, in some following experiments, we did not add and repeat to compare the WT and LOXL3 knockdown cells as control, just comparing the WT and LOXL3 mutant restoration, for less waste and consumption, better data presentation, and the ethics of reducing animal usage as possible. For example, in **Fig. 1(g, l)**, if we added the WT and LOXL3 knockdown cells in the cell viability curve, it would be very hard for people to Fig. out so many curves and their difference and impact the understanding of the study, plus with the fact that their viability curve already existed in Supplemental Fig. 1F. Likewise, there is the similar reason for **Fig. 3k**, **Fig. 4k**. Especially, for the animal xenograft experiment, in **Fig. 5a-d**, based on the solid in vitro phenotype of cell lines and in vivo phenotype of site-mutation *Loxl3-S704D* mice as well as the ethics of reducing animal usage, we did not include the WT and LOXL3 knockdown cells in the xenograft experiment which would not affect reliability of our conclusion.

2. *Fig. 1J, K, L: "we constructed cell lines stably expressing wild-type or enzymatically dead (ED) forms of LOXL3 with mitochondrial signal peptides (named M-WT-L3 or M-ED-L3) in LOXL3-deficient cells (Fig. S1G-H)." Since LOXL3 can localize in mitochondria, what is the point to add a mitochondrial signal peptide to LOXL3? It seems that their data in Fig. 1J, K did not prove anything, since the effect of expressing WT-L3 and M-WT-L3 is similar. Also, they did not include ED-L3 here, and there is no comparison for M-ED-L3. The authors need to compare mitochondrial localization (by fractionation as shown in Fig. 1H) of WT-L3 and M-WT-L3 (as well as ED-L3 and M-ED-L3). I assume LOXL3 does not have the classic*

mitochondrial localization signal, so adding a mitochondrial localization signal to LOXL3 should increase its localization into mitochondria. If so, why there is no functional difference between WT-L3 and M-WT-L3? What does the story gain by showing this data?

Response: Thanks for the reviewer's mention, we have added the experiment group of ED-L3 restoration cells in **Fig. 1j-k** and supplemented the fractionation of WT-L3 and M-WT-L3 (as well as ED-L3 and M-ED-L3) in **Supplemental Fig. S1r** to compare mitochondrial localization.

In our opinion, we agree with the reviewer's point that effect of expressing WT-L3 and M-WT-L3 was similar, but not the point that data in Fig. 1J, K did not prove anything. As LOXL3 partly located in mitochondrial, the fusion of mitochondrial signal peptide to LOXL3 should expectedly reinforce LOXL3 location in mitochondrial and the effect of mitochondrial LOXL3 on anti-ferroptosis. In fact, unexpectedly, though it was true that adding a mitochondrial localization signal to LOXL3 increased its localization into mitochondria (**Supplemental Fig. S1r**), M-WT-L3 just showed a slight stronger capacity for anti-ferroptosis than WT-L3 (**Fig. 1j-k**), which implied the information that the upstream regulation of LOXL3 may be more important. We speculate that the upstream EGFR signaling not only promotes the interaction of LOXL3 and TOM20 that pulls LOXL3 into mitochondrial, but also change the conformation of LOXL3 by protein modification, mediating the recognition and phosphorylation of LOXL3 by AK2 in mitochondrial. We will verify the hypothesis and deeply study the molecular regulation mechanism in the further study of future.

3. Protein levels of LOXL3 and DHODH in cell lines and experimental conditions in Fig. 3J need to be shown. (There is a misspelling in Fig. 3J "DHODH-Myc".)

Response: Thanks a lot for the reviewer's mention and suggestion. We immunoblotted the protein levels of LOXL3 and DHODH in indicated cells by western blot and have added the requested data in **Fig. 3j**.

4. Is ferroptosis at least partly responsible for the decreased cell viability and increased cell death in LOXL3 knock-down cells with Oxa treatment (Fig. 1 and Fig. S1C)? This can be addressed by examining whether adding the ferroptosis inhibitor ferrostatin-1 or liproxstatin-1 can rescue cell death and restore cell viability in this context.

Response: Thanks a lot for the reviewer's constructive suggestion. According to his or her suggestion, we added the ferroptosis inhibitor ferrostatin-1 to the LOXL3 knock-down cells with Oxaliplatin treatment. The result that ferrostatin-1 blocked the effect of LOXL3 knockdown with Oxaliplatin treatment on cell viability and cell death implied the fact that ferroptosis was mainly responsible for the decreased cell viability and increased cell death in LOXL3 knock-down cells with Oxaliplatin treatment (**Supplemental Fig. S1k-l**).

5. Fig. 1, Fig. 4F: the authors need to confirm their major findings with at least two si/shRNAs for LOXL3 and AK2.

Response: Thanks a lot for the reviewer's mention and suggestion. According to his or her

suggestion, we used two shRNAs targeting to LOXL3(Supplemental Fig. S1e-g) and two siRNAs targeting to AK2(Fig. 4c-d, f) to confirm our major findings.

6. Fig. 5G: under Oxa treatment condition, the levels of DHODH in WT and S704D samples are the same. If so, how to explain the difference in lipid peroxidation and tumor phenotypes between WT and S704D mice under Oxa treatment (Fig. 5H-K)?

Response: Thanks a lot for the reviewer's mention. As we all known, it is a process that cells abnormally response to signals and stress that eventually resulting in tumor progression and resistance to chemotherapy. Our data showed that DHODH protein level would be elevated in response to Oxaliplatin treatment for defending the stress. Although DHODH expression level of WT mice caught up with LOXL3-S704D mice after treatment with Oxaliplatin at 6th week, the degree of response to Oxaliplatin was different from the beginning of the front because at the beginning, the liver from S704D mice contains more DHODH which conferred the S704D mice with more resistance to Oxaliplatin induced Lipid peroxidation. Once the HCC cells response to the Oxaliplatin treatment, after accumulation for 6 weeks, the liver from S704D mice were much more resistant to Oxaliplatin induced Lipid peroxidation than wild type, leading to the tumor phenotypes between WT and S704D mice under Oxa treatment.

Minor comments:

1. The manuscript writing needs to be significantly improved. Suggest the authors to seek help from a professional writing editor:

Response: Thanks a lot for the reviewer's mention and constructive suggestion. To improve the language, we sent this manuscript out to be edited by specialists in Oncology and native English speakers which were suggested by NPJ group.

2. Also, in the Introduction, they mixed background introduction with their findings, which is confusing (in Introduction of most other papers, the authors first introduce relevant background and highlight unmet needs and knowledge gaps, and briefly summarize their findings in the last paragraph of Introduction). Introduction needs to be re-drafted.

Response: Thanks a lot for the reviewer's mention and constructive suggestion. We rewrote introduction, discussion and simply described the results.

3. It is really painful to examine their clonogenic survival data, which contain multiple lines with similar colors. The authors can add other features (such as solid and dash lines) to distinguish different groups.

Response: Thanks a lot for the reviewer's mention and constructive suggestion. We have added lines to link the important cell groups needed to be highlighted and compared, for drawing the conclusion. We hope that it is now clearer to examine our survival data and the comparisons of important cell groups.

Reviewer 2#:

The proposed manuscript of Wang et al. investigated the molecular mechanisms of drug resistance in hepatocellular carcinoma (HCC) driven by Lysyl oxidase-like 3 protein (LOXL3) and its regulatory elements. The authors propose that resistance to Oxaliplatin is initiated by EGFR signaling and propagated by AK2-LOXL3-DHODH axis in mitochondria which ultimately affects the rate of ferroptosis. They identified that LOXL3 exerted high activity in mitochondria and is regulated by AK2 kinase. Once activated, LOXL3 prevents ubiquitination of DHODH which consequently decreases mitochondrial ferroptosis and displays higher resistance to chemotherapy. Drug resistance represents an increasing obstacle for patients with HCC, therefore this study focuses on a timely and relevant topic. The project is technically well executed and provides novel mechanistic insights into drug resistance in HCC. Utilization of the cell lines, animal models, and human samples provided potential translational value.

However, there are several questions and open issues that limit the enthusiasm for the manuscript. Further, several issues related to the selection of the drug, exploration of underlying mechanism as well as lack of independent external validation exist.

Response: We thank a lot for the reviewer's positive comments, insightful criticism and constructive suggestions which heavily strengthen our study. As such, we addressed the questions raised by the reviewer and revised the manuscript to improve the clarity.

Main Comments:

1. The selection of platin-based compounds for the evaluation of liver cancer drug resistance is unclear. It should be clearly explained and demonstrated if LOXL3 exerts general properties of chemoresistance or if a platin-dependent effect is proposed. Herein, given that chemotherapy plays a minor role in HCC treatment it would be beneficial to validate results with approved drugs for HCC in the context of LOXL3. In line with this, Sorafenib is reported to induce autophagy, apoptosis, and activates ferroptosis, which might make this compound particularly promising.

Overall, the rationalization of the Oxaliplatin investigation as well as the focus on liver cancer should be delineated in more detail and the rationale should be provided for the actual human diseases.

Response: Thank a lot for the reviewer's mention. We apologize for not clearly explaining the reason why we studied the chemoresistance of Oxaliplatin in the treatment of liver cancer in the initial manuscript. Our explanation of the rationalization of the Oxaliplatin investigation as well as the focus on liver cancer is following:

Approximately 25-70% of HCC patients are diagnosed at an advanced stage, whose median overall survival (OS) is only 4.2-7.9 months and lack of treatment options. To date, there is still few first-line approved treatment, such as Sorafenib or Lenvatinib, shown to extend OS for advanced hepatocellular carcinoma (PMID: 28983565, PMID: 19095497, PMID: 18650514, PMID: 26795574). However, limitations heavily affect the use of approved treatment in advanced hepatocellular carcinoma, including modest survival advantage, low response rates and so on (PMID: 19095497, PMID: 18650514, PMID: 26170167, PMID:

28045619). So, there exists the urgent requirement of other more alternative therapies for advanced hepatocellular carcinoma.

FOLFOX (**Oxaliplatin plus fluorouracil and leucovorin**) was a regimen first used in colorectal cancer with liver metastases and reported to be effective both by systemic and HAIC in clinical trials (PMID: 25448804, PMID: 28426374). Fortunately, HAIC of FOLFOX (FOLFOX-HAIC) was well tolerated and effective in hepatocellular carcinoma, improving the survival benefits compared to sorafenib, which was partially supported by a recent prospective randomized trial (PMID: 28592441, PMID: 29471013, PMID: 31070690, PMID: 31070690).

But as for the future role of FOLFOX-HAIC in hepatocellular carcinoma treatment, it is more likely to just serve as a supplemented method of reducing tumor burden while preserving hepatic arterial blood supply to the tumor, since chemotherapy resistance can develop after multiple sessions of FOLFOX-HAIC. Besides, for reaching the effective concentration of chemotherapeutic agents to kill tumor cells, the receiving dosage of chemotherapeutic agents for patients in vivo remains too high, resulting in high side-effect and toxicity.

Herein, we aim to overcome the chemotherapy resistance and toxicity developed by FOLFOX to enhance the efficacy of FOLFOX-HAIC in hepatocellular carcinoma treatment and eventually benefit the HCC patients. There was the reason why we went back to study the chemoresistance of in the treatment of liver cancer. In this paper, we mainly studied the effect of LOXL3 on HCC cells responding to Oxaliplatin.

In order to give a complete account of LOXL3 in providing tolerance to chemotherapeutic agents, we supplemented the data about the effect of 5-Fu on hepatocellular carcinoma cells with or without LOXL3 in **Supplemental Fig. S1c**, which implied LOXL3 exerted general properties of chemoresistance, without a platin-dependent manner.

2. The authors mention that Oxaliplatin causes massive cell death in shLOXL3 cells which prevents detailed investigation of ferroptosis. Given that viability is generally over 50% (Fig. 1), this is difficult to understand and should be explained.

Response: Thank a lot for the reviewer's mention. We apologize to not clearly describe this sentence. Ferroptosis is an intracellular iron-dependent form of programmed cell death that is distinct from apoptosis, necrosis, and autophagy with respect to associated genetic processes, biochemical activities, and morphological characteristics, resulting from unrestrained lipid peroxidation (PMID: 17568748, 22632970). Efforts to increase cell susceptibility to ferroptosis sensitized cancer cells to chemotherapy and reduced the drug resistance, implying the role of ferroptosis in chemotherapy resistance (PMID: 35151318, PMID: 31101865). However, different from the ferroptosis inducer Erastin, it was hard to obtain clear ultrastructural evidence of ferroptosis when focused on the chemotherapy induced ferroptosis, because it was disturbed by morphological characteristics of apoptotic cells.

Therefore, we chose the time point that the HCC cells were treated with Oxaliplatin just for 6 hours at which time point, the extremely minor cells undergo apoptosis. Fortunately, we obtained the clear ultrastructural evidence of ferroptosis in shLOXL3 cells before obvious apoptotic morphology. Of note, we mentioned in the original manuscript that Oxaliplatin caused massive cell death in shLOXL3 cells preventing detailed investigation of ferroptosis. Though the cell viability was generally over 50% in Fig. 1, the apoptotic morphological events already

widespread took place in cells.

3. The mechanism of how EGF/EGFR activation regulates LOXL3 remains uncertain and should be experimentally addressed. How does EGFR activate TOM20 protein and initiate translocation of LOXL3 inside mitochondria?

Response: Thank a lot for the reviewer's mention and his or her constructive suggestions. It is interesting that, as LOXL3 partly located in mitochondrial, the fusion of mitochondrial signal peptide to LOXL3 should expectedly reinforce LOXL3 location in mitochondrial and the effect of mitochondrial LOXL3 on anti-ferroptosis. But, in fact, M-WT-L3 (WT-LOXL3 with mitochondrial signal peptide) just showed a slight stronger capacity for anti-ferroptosis than WT-LOXL3 (**Fig. 1j-k**), though adding a mitochondrial localization signal to LOXL3 truly increased its localization into mitochondria (**Supplemental Fig. S1p**).

We speculate that the upstream EGFR signaling not only promotes the interaction of LOXL3 and TOM20 that pulls LOXL3 into mitochondrial, but also changes the conformation of LOXL3 by protein modification, mediating the recognition and phosphorylation of LOXL3 by AK2 in mitochondrial. Additionally, the possibility of directed recruitment by EGFR cannot be excluded either. Moreover, the exact protein site of adaptor proteins responsible for regulation should also be identified and the manner how the site response to EGFR activation, including those experiments about reversed genetics and restoration. Overall, it is hard to verify the hypothesis and deeply study the molecular regulation mechanism in the study as the length of this article is already long and we mainly focused on the axis of AK2/LOXL3/DHODH. We apologize to the reviewer for not conducting those experiments to explore the molecular mechanism of EGFR/LOXL3/TOM20 axis formation in depth at this occasion. It is a truly very important and constructive comment, we will explore the mechanism clearly in the further study of future.

Despite the fact above, we still supplemented experiments to further confirm the regulation of EGFR signaling on AK2/LOXL3 axis. We treated HCC cells with a combination of low-dose Oxaliplatin and EGFR mono-antibody Cetuximab. As a result, in shNT group, but not shLOXL3 group, Cetuximab significantly increased the lipid peroxidation and cell death when combined with Oxaliplatin (**Supplemental Fig. S2a**). The combination of Oxaliplatin and Lenvatinib has fewer efficiency than the combination of Oxaliplatin and Cetuximab in shNT group, but more efficiency in shLOXL3 group. In addition, when we combined Oxaliplatin, Cetuximab with Lenvatinib, comparing to the combination of Oxaliplatin and Lenvatinib, the lipid peroxidation increased in shNT group but not in shLOXL3 group (**Supplemental Fig. S2b**). Above data showed that EGFR-driven the blockage of lipid peroxidation depends on LOXL3, and the combination of Oxaliplatin and Lenvatinib works better in the absence of LOXL3. Also, in **Fig. 4d**, we treated the cells with EGF after starvation overnight, the phosphorylation of LOXL3-S704 was heavily activated, but abolished by AK2 knockdown. These data further strengthened the axis of EGFR/LOXL3/AK2 and its regulation function in ferroptosis and chemotherapy resistance.

4. If EGFR-driven chemoresistance to Oxaliplatin is the key molecular mechanism, can this be validated by inhibition of EGF/EGFR pathway? Can results of LOXL3 be recapitulated by

combining EGFR inhibitor and Oxaliplatin, i.e., increased level of ferroptosis and increased cell death. Moreover, tyrosine kinase inhibitors that are approved for HCC (e.g., Sorafenib, Lenvatinib), affect EGFR signaling which should be explored in the context of the study.

Response: Thank a lot for the reviewer's mention and his or her constructive suggestion.

In new **Supplemental Fig. S2a-b**, we treated HCC cells with a combination of low-dose Oxaliplatin, Cetuximab (EGFR mono-antibody), or Lenvatinib. As a result, in shNT group but not shLOXL3, Cetuximab significantly increased the lipid peroxidation and cell death when it was combined with Oxaliplatin (**the fourth group of Supplemental Fig. S2b-b**). The combination of Oxaliplatin and Lenvatinib (**the fifth group of Supplemental Fig. S2a-b**) has fewer efficiency than the combination of Oxaliplatin and Cetuximab (**fourth group**) in shNT group, but more efficiency in shLOXL3 group. In addition, when we combined Oxaliplatin, Cetuximab with Lenvatinib (**the sixth group of Supplemental Fig. S2a-b**), comparing to the combination of Oxaliplatin and Lenvatinib (**fifth group**), the lipid peroxidation increased in shNT group but not in shLOXL3 group. Above data showed that EGFR-driven the blockage of lipid peroxidation depends on LOXL3, and the combination of Oxaliplatin and Lenvatinib works better in the absence of LOXL3.

5. To confirm the role of ferroptosis in this process, it would be interesting to assess direct activation of ferroptosis rather than solely using indirect activation by chemotherapeutic compound.

Response: Thank a lot for the reviewer's mention and constructive suggestion. In new **Supplemental Fig. S1m-n**, Erastin, a specific inducer of ferroptosis, was used to directly activate ferroptosis to confirm the role of LOXL3 in ferroptosis. As we seem, the lipid peroxidation was significantly increased for the lack of LOXL3, not only in the Oxaliplatin treated group which we confirmed in previous data, but also in the group treated by Erastin. Additionally, we found the synergized effect was enhanced in the LOXL3 knockdown cells, which further implied the role of LOXL3 to interpret chemotherapy response by ferroptosis.

6. The relevance of the suggested findings for human HCC is interesting. The investigated cohort should be described in more detail and clinico-pathological information for the described patients (Fig. 7) should be provided. Can the authors confirm an association to response to the therapy, e.g., histologically, depending on the expression of AK/LOXL3/DHODH?

Response: Thank a lot for the reviewer's positive comment and his or her constructive suggestion. We have added the clinico-pathological information about the patients in the cohort to the **Supplemental Table 1**. Furthermore, we think the result of PDX model, PDX xenografts with high level of pLOXL3-S704/DHODH were more resistance to Oxaliplatin treatment, could confirm the association the response to the therapy with AK/LOXL3/DHODH axis. Additionally, we supplemented the PFS (Progression Free Survival) analysis of the patients receiving chemotherapy treatment in the cohort, respectively using the AK/LOXL3/DHODH

histologically expression (**Supplemental Fig. S7e**).

7. The suggested prognostic implication is interesting. Independent validation in the external dataset would be interesting. In addition, predictive impact for therapy other than Oxaliplatin should be demonstrated.

Response: Thank a lot for the reviewer's positive comment, it's grateful that we feel sorry for this suggestion. Since the significance of the prognostic value on chemotherapy response was depended on LOXL3-S704 phosphorylation, relied on the specific LOXL3-S704 phosphorylation antibody, but not the overall protein or mRNA levels of LOXL3. However, the public clinical database, such as TCGA, Oncomince, cBioportal or other published datasets on GSE from other papers, mainly just offer the mRNA expression using next-generation sequencing technology. By mining TCGA-LIHC cohort, we showed the clinical significance of LOXL3 mRNA in **Supplementary Fig. S7a and f**, which also supported the oncogenic role of LOXL3.

As we focused on the clinical significance from the protein translational modification of LOXL3, we cannot validate the prognostic value of LOXL3-S704 phosphorylation on chemotherapy response in other external datasets with mRNA expression. For the same reason, patients here all receiving the regimen FOXFOL4 for HCC treatment, both containing Oxaliplatin and 5-Fu. Thus, it is hard for us to predictive impact for therapy other than Oxaliplatin. Despite all this, the data about the effect of 5-Fu on HCC cells with or without LOXL3 in **Supplemental Fig. S1c** could partially offer positive clue for predictive impact for therapy other than Oxaliplatin, like 5-Fu. Based on the main aim to improve the therapy FOLFOX-HAIC regimen, in our option, it can be accepted the specification for Oxaliplatin and 5-Fu.

8. The validation of the findings in PDX is interesting and relevant. The clinic-pathological information should be provided and the investigations should be described in more details. In particular, it is unclear how many tumor specimens were processed and how the two groups were established.

Response: Thank a lot for the reviewer's positive comment. According to the reviewer's suggested, we supplemented the clinic-pathological information of the patients offering the tumor tissues for establishing the PDX model. The process of our PDX models have been describe in detail into our method in the part of experimental mice. The two groups were established by histologically examination, with further verified in PDX experiments (**Supplemental Fig. S7g-h**).

9. Consistency in data representation is unclear (the result section mainly presents data mostly for Hep3B cell line). The results should be consistently presented throughout for at least both of the cell lines.

Response: Thanks for the reviewer's mention and conductive suggestion. We supplemented these data using both Hep3B and Huh7 cells in this study (**Supplemental Fig. S1(e-g, i-j, q-p)**),

Fig. 5i, Supplemental Fig. S5e) to keep consistency in data representation, especially at the first time to identify the phenotype in HCC cells. However, based on the consistent phenotypes already confirmed both in Huh7 and Hep3B cells and the research conventions, we mostly used Huh7 to explore how the EGFR/LOXL3/AK2/DHODH axis regulated ferroptosis to restrain the chemotherapy response. Moreover, we verified key findings in Hep3B cells to keep the data consistency in our study, like those functional experiments that phosphorylation of LOXL3-S704 regulated the LOXL3 activity (**Fig. 2l**), phosphorylation of LOXL3-S704 regulated ferroptosis mediated by DHODH (**Fig. 3k**), LOXL3 regulated ferroptosis mediated by the phosphorylation by AK2(**Fig. 4i-k**) and so on. Additionally, we had generated LOXL3 site mutation mice and verified our findings in vivo using the mutant mice, strongly supporting our findings conserved in HCC chemotherapy response.

Minor comments:

1.The provided abstract requires streamlining and simplification for readers, not familiar with the topic.

Response: Thanks for the reviewer's mention and constructive suggestion. We have improved our manuscript including abstract, introduction and discussion part, with the more simplified result description. To improve the language, we sent this manuscript out to be edited by specialists in Oncology and native English speakers.

2.Fig. 6l shows no difference between the Leflunomide and combined Leflunomide/Oxaliplatin. It would be expected that DHODH activity would be further decreased when used in combination. Please comment.

Response: Thanks for the reviewer's question. We comment that the result of DHODH activity in this experiment is reasonable. Leflunomide is an effective DHODH activity inhibitor, which could block the DHODH activity whether treated with Oxaliplatin or not, resulting in the same level of DHODH activity between the Leflunomide and combined Leflunomide/Oxaliplatin treatment group. In the combined Leflunomide/Oxaliplatin treatment group, the additional tumor killing ability was due to the loss of DHODH in defending ferroptosis induced by Oxaliplatin and the toxicity of chemo-drugs, but not the further decreased DHODH activity.

3.Results from the animal model in Fig. S6A - the effects between Oxaliplatin and LOXL3 inhibitors look additive rather than synergistic. Following the claims of the authors, synergism is expected. Please elute.

Response: Thanks for the reviewer's mention. We speculate that the reason why synergism did not occur in the in vivo experiment may be due to the effect on the tumor microenvironment, especially the immunology cells. In the combined drug treatment group, the tumor cells trended to massively die which may be pushed to elevate and secrete several cytokines to recruit some tumor supported immune cells like tumor associated macrophages (TAMs) or neutrophils (TNs), which defend the synergism resulting in the additive effect.

3. What is the reason to use female mice for xenograft studies and male mice for HTVI-induced HCCs

Response: The reason for selecting male mice for HTVI-induced HCCs is that, generally, HCC model should be more efficiently established in male mice. Xenograft studies require avoidance of some effects from the host mouse on tumor cell whenever possible.

Reviewer 3#:

In this manuscript, Zhan et al report that lysyl oxidase-like 3 (LOXL3), but not other members of LOX family, confers resistance to Oxaliplatin in liver cancer cell lines. They showed that upon depletion of LOXL3, liver cancer cell lines become responsive to Oxaliplatin by increasing lipid peroxidation and inducing ferroptosis. The role of LOXL3 in Oxaliplatin resistance in vitro has been validated using a robust in vitro add-back system where authors stably knocked down LOXL3 and overexpressed different versions of LOXL3, including wild type, enzymatically dead, mitochondria-directed ones, etc. They showed that EGF induces the localization of LOXL3 in mitochondria by inducing its interaction with TOM20, a member of mitochondrial outer membrane. The phosphorylation and activation of mitochondrial LOXL3 was shown to be mediated by mitochondrial adenylate kinase 2 (AK2) at S704 site, and this phosphorylation was shown to confer resistance to chemotherapy-induced ferroptosis. The authors also demonstrated that LOXL3 phosphorylation and activation prevent ubiquitination of DHODH (a key mitochondrial protein involved in several metabolic process) by oxidizing the lysine residues on DHODH leading its stability, which ultimately results in resistance to Oxaliplatin -induced ferroptosis. Although not clear, authors also showed S704D mutation (inducing LOXL3 homodimerization) on LOXL3 confers chemoresistance in a liver tumor model. In addition, using xenografts and PDXs, they examined the contribution of S704A mutation on Oxaliplatin resistance in vivo. Finally, they tested DHODH inhibitor together with Oxaliplatin in a liver tumor model and examined the expression of AK2/pLOXL3/DHODH axis in liver patients' tissues by immunohistochemistry.

While the manuscript reports an interesting non-canonical function of LOXL3 where mitochondrial LOXL3 and its mitochondrial-specific phosphorylation plays role in ferroptosis and potentially in chemoresistance, there are major concerns which need to be addressed.

Response: We thank a lot for the reviewer's positive comments, insightful criticism and constructive suggestions which heavily strengthen our study. As such, we performed requested experiments by the reviewer and revised the manuscript to improve the clarity.

Main Comments:

1. It is not clear why authors have chosen two liver cancer cell lines (Huh7 and Hep3B) for this study. Are they resistant to Oxaliplatin, which is the major chemotherapy used in this study? Furthermore, it is not clear why Oxaliplatin, but not other chemotherapy agents used in liver cancer, is chosen throughout the study. If the authors claim that LOXL3 is critical for "chemoresistance" in general, they should repeat some of the experiments with other

chemotherapy agents.

Response: We chose Huh7 and Hep3B cells as they are two representative HCC cell lines for study chemotherapy response in liver cancer. Chemo-resistance generally refers to the decreased sensitivity of drug response after repeated cycles of chemotherapy, resulting in the reduction or invalidity of the efficacy. Besides, for reaching the effective concentration of chemotherapeutic agents to kill tumor cells, the receiving dosage of chemotherapeutic agents for patients in vivo remains too high, resulting high side-effect and toxicity. Thus, efforts to increase the sensitivity of chemotherapy response would not only decrease the chemoresistance, but also alleviate the heavy toxicity induced by a dosage of chemotherapy. Overall, our study here was aimed to offer an alternative way to increase the sensitivity of HCC tumors response to the chemotherapeutic agents for better efficacy and lower toxicity, and Huh7 and Hep3B cells are two representative HCC cell lines for studying the function of drug resistance, tumor formation and so on.

As for the choose of Oxaliplatin, we decided it based on the chemotherapy regimen FOXFOL4 for hepatocellular carcinoma (HCC) and FOXFOL4 was reported to be effective both by systemic and hepatic arterial infusion chemotherapy (HAIC) in clinical trials (**PMID: 25448804, PMID: 28426374**). HAIC of FOLFOX (FOLFOX-HAIC) was well tolerated and effective in hepatocellular carcinoma, improving the survival benefits compared to sorafenib, which was supported by a prospective randomized clinical trial (**PMID: 28592441, PMID: 29471013, PMID: 31070690, PMID: 31070690**). Oxaliplatin and 5-Fu, as the traditional chemotherapeutic agents for cancer, have the problem of great toxicity and drug resistance with low drug response, especially Oxaliplatin (**PMID: 30982686, PMID: 28542671, PMID: 33128031**). Thus, we focused on Oxaliplatin treatment at first.

Next, in order to give a complete and general account of LOXL3 in providing tolerance to chemotherapeutic agents, we supplemented the data about the effect of 5-Fu on hepatocellular carcinoma cells with or without LOXL3 in **Supplemental Fig. S1c**, which implied LOXL3 exerted general properties of chemoresistance, without a platin-dependent manner. Our aim is to improve the efficacy of FOLFOX (Oxaliplatin plus 5-Fu and leucovorin) regimen for patients with low side-effect and provide the reference for more clinical trials and effective therapies in HCC treatment.

2.

1) While the authors claim that chemotherapy-induced EGFR signaling is responsible for the mitochondrial LOXL3 expression, it is not shown in this paper. This is critical as the upstream of the LOXL3 activation is shown to be via EGF signaling. In it is current form, there is a major disconnect from Oxaliplatin to EGFR activation to mitochondrial LOXL3 induction.

Response: Thanks for the reviewer's mention. We apologize to not describe clearly. But we cannot agree with the reviewer's option that there is a major disconnect from Oxaliplatin to EGFR activation to mitochondrial LOXL3 induction. Actually, it was fully confirmed in our study that EGF/EGFR signaling was responsible for the mitochondrial LOXL3 activation and its role in anti-ferroptosis induced by Oxaliplatin to promote chemoresistance, mainly supported by the result that the restored expression of LOXL3 K35/36A mutant, the site

mediating the mitochondrial entry of LOXL3 induced by EGF/EGFR signaling activation, could not rescue the lipid peroxidation and cell death in LOXL3 knockdown HCC cells, which we will further illustrate clearly in the next concern about the same issue.

As for the connection of chemotherapy and EGFR signaling, it is well known that chemotherapies cause EGFR signaling pathway activation to promote chemoresistance, which was correlated with our findings in the study (PMID: 24295852, PMID: 22157681, PMID: 23242808, PMID: 21741919). For example, Oxaliplatin treatment could not only increase the LOXL3's entry into mitochondria which was proved to be mediated by EGF/EGFR signaling activation (Fig. 1h-i, 2a-g), but also the LOXL3-S704 phosphorylation by AK2 in mitochondrial and the consequent DHODH protein level (Fig. 2j, 3b).

Furthermore, we supplemented the experiment result of treating HCC cells with a combination of Oxaliplatin and EGFR mono-antibody, Cetuximab. As a result, in shNT group, but not shLOXL3 group, Cetuximab significantly increased the lipid peroxidation and cell death when combined with Oxaliplatin (Supplemental Fig. S2a), which implied the activation of EGFR signaling pathway in Oxaliplatin treatment and its role in promoting chemoresistance by activation of LOXL3 to defend Oxaliplatin induced ferroptosis.

Another direct evidence showed Cetuximab mediated blockade of EGFR signaling would diminish Oxaliplatin induced upregulation of LOXL3-S704 phosphorylation, which only occurred in mitochondria (Supplementary Fig. S2j).

2) Furthermore, it is not clear if Oxaliplatin changes mitochondrial LOXL3 activity (compare Supp Fig 1J and L). This needs to be clarified.

Response: Thanks for the reviewer's mention. Oxaliplatin treatment induced LOXL3 translocation from cytosol into mitochondria (Fig. 1h-i), where LOXL3 was not only phosphorylated by AK2 (Fig. 3), but also regulated DHODH protein level to defend the ferroptosis induced by Oxaliplatin treatment (Fig. 4). We proved that the phosphorylation of LOXL3-S704 by AK2 in mitochondrial was critical for LOXL3 activity (Fig. 3l), as well as, the fact that restoration of LOXL3 S704A mutant could not rescue the phenotype (Supplemental Fig. S3f) which means the phosphorylation of LOXL3-S704 by AK2 was critical for LOXL3 function and activity. So, we preliminarily concluded that enhanced LOXL3-S704 phosphorylation, upon Oxaliplatin treatment, has higher activity to defend the ferroptosis induced by Oxaliplatin.

To further directly confirm the elevated activity of LOXL3 in mitochondria, cytosol or mitochondrial LOXL3-FLAG were respectively enriched and purified from Huh7 or Hep3B cells stably expressing LOXL3-FLAG, with treatment of Oxaliplatin for half hour. Then, LOXL3-FLAG was measured for activity in vitro. Eventually, it presented higher mitochondrial LOXL3 activity response to Oxaliplatin treatment (Supplemental Fig. S1t).

3. Another disconnect is the identification of AK2 as the mitochondrial kinase phosphorylating LOXL3 on S704 site. Although AK2 phosphorylation of LOXL3 is convincing, it is not clear if LOXL3 phosphorylation by AK2 is EGF dependent. In other words, is its phosphorylation by AK2 mediated by chemotherapy-induced EGFR activation? If not, it is hard to claim the axis presented here as responsible mechanism for mitochondrial LOXL3 mediated chemoresistance.

Response: Thanks so much for the reviewer's mention. In our study, we found that: 1) EGF/EGFR signaling was required for LOXL3 binding with TOM20, whose interaction was mediated by K35/36 site of LOXL3 (**Fig. 2c, 2f**). Meanwhile, the site mutation of K35/36A blocked the LOXL3 entry into mitochondrial (**Fig. 2g**) and the anti-ferroptosis ability to promote chemotherapy resistance (**Fig. 2h-i**). So, it was concluded that the interaction of LOXL3 with TOM20 or entry into mitochondrial of LOXL3 was dependent on the EGF/EGFR activation. 2) It was further identified that mitochondrial LOXL3 was phosphorylated by AK2, a kinase that was specific in mitochondrial (**Fig. 4**).

For the direct evidence, we furtherly treated the cells with EGF after starvation overnight by removing serum in the medium. It showed that the phosphorylation of LOXL3-S704 phosphorylation was heavily low in the untreated with EGF cells, which was highly activated after EGF treatment, but blocked by AK2 knockdown (**Fig. 4d**). Further, we used low-dose Oxaliplatin to treat cancer cells while Cetuximab were applied to block EGFR signaling. The results showed the blockade of EGFR signaling by Cetuximab would diminish Oxaliplatin induced upregulation of LOXL3-S704 phosphorylation (**Supplementary Fig. S2j**).

Overall, based on the key finding above, we could confirm the conclusion that LOXL3 phosphorylation by AK2 at S704 was EGF/EGFR signaling activation dependent, further strengthening the EGFR/LOXL3/AK2 axis in chemotherapy resistance by ferroptosis regulation.

4. DHODH is a key protein playing roles in both pyrimidine biosynthesis and mitochondrial respiratory chain. Although the study assumes DHODH as proxy for the lipid peroxidation and ferroptosis, it is not clear what DHODH stabilization upon LOXL3 activation does on these other key metabolic processes which are also potential major mechanisms involved in chemoresistance. This needs to be addressed experimentally.

Response: Thanks so much for the reviewer's mention and constructive suggestions. In new Fig. S3I-K, to clarify the redox function or pyrimidine biosynthesis of DHODH in contributing to ferroptosis in our study, in S704A cells (down-regulated DHODH), we measured uridine level and found the down-regulation of uridine in S704A cells. However, the supplementation of uridine did not enhance Oxaliplatin resistance of HCC LOXL3-S704A cells (**Supplemental Fig. S3i-k**).

5. Also, while the authors measured the cytoplasmic ROS and did not observe a major change upon LOXL3 modulation, what happens to mitochondrial ROS which is also controlled by DHODH in mitochondria?

Response: Thanks for the reviewer's mention. In this study, at first, we observed the massive death of LOXL3 knockdown HCC cells under Oxaliplatin treatment. Based on the more and more important role of ROS and ferroptosis in regulating cell death and drug resistance, we next measured the cellular ROS and lipid peroxidation level of the cells. Surprisingly, under Oxaliplatin treatment, a significant elevation of lipid peroxidation was observed in LOXL3-deficient cells while LOXL3 deficiency only slightly altered cellular ROS.

Mitochondria also plays a key role in lipid peroxidation and ferroptosis (**PMID:**

30581146). Combined with the evidence that mitochondrial localization of LOXL3 and the higher sensitivity of the HCC cells with LOXL3 depletion under Oxaliplatin treatment, thus, we speculated that the significant elevation of lipid peroxidation of LOXL3 deficiency upon Oxaliplatin treatment was mainly caused by mitochondrial accumulated lipid ROS, resulting in ferroptosis and consequent cell death (**Fig. 1**). Therefore, it prompted us to start the study the possibility of LOXL3 in mitochondria and its corresponding mechanism to regulate ferroptosis.

As for mitochondria ROS the reviewer mentioned, we think it should be consistent to the accumulated lipid ROS in mitochondria since the concept lipid ROS means the process of ROS oxidizing polyunsaturated fatty acids. **Additionally, it should be noted here that we measured the cellular ROS by H2DCFDA**, a unique cell-permeable fluorogenic probe (BioVision, Waltham, MA; #K936-100). Upon the cell entry, H2DCFDA is modified by cellular esterases to form a non-fluorescent H2DCF. Oxidation of H2DCF by intracellular ROS yields highly a fluorescent product that can be detected by fluorescence microscope, whose intensity is proportional to the ROS levels. But the non-fluorescent H2DCF has not the permeable ability and will be retained in cytoplasm, specifically sensing cytosolic ROS (**PMID: 10443931**). For the reliable lipid peroxidation phenotype of ferroptosis that further strengthened by treated with ferroptosis inhibitor, ferrostatin-1(**Supplemental Fig. S1k-l**), and the data presentation length limitation in our manuscript, we did not add the result of measuring mitochondrial ROS in our manuscript. Here, we utilized the commercialized probe, MitoSOX (Invitrogen, M36008), to measure the mitochondrial ROS (Ex/Em 396/610 nm). The result is following:

Explanation to the result is that, though DHODH was downregulated in LOXL3 knockdown cells, the cells had strong anti-oxidative system to balance the loss of DHODH to restrict the ROS elevation and protect the cells. But, under Oxaliplatin treatment, the anti-oxidative system was faced with much strong oxidative pressure. The loss of DHODH accelerated imbalance of the redox system, resulting in significant elevation mitochondrial ROS.

6. Overall, the in vivo experimental set-ups are not clear at all, and the results section describing those findings are not written well. Indeed, the in vivo experiments are missing the validation of key in vitro findings. Importantly, testing DHODH inhibitor together with Oxaliplatin is again missing the context of the manuscript with respect to LOXL3 involvement. A better experimental design is needed to show that DHODH is a key contributor to LOXL3-driven chemoresistance in vivo. Furthermore, treatment schemes, treatment duration, doses, and sample sizes are confusing across the board.

Response: Thanks for the reviewer's mention and his or her constructive suggestion. According to the reviewer's mention, we rewrote our in vivo experiment results describing the findings

more clearly. According to the reviewer's suggestion about LOXL3 involvement and dependence on DHODH, the LOXL3-S704D mutant mice, mimicking LOXL3 activation and more resistant to Oxaliplatin treatment, were used in the experiment of testing DHODH inhibitor together with Oxaliplatin. As shown in **Supplemental Fig. S6b-f**, under the treatment of Oxaliplatin, the combination of DHODH inhibitor Leflunomide would efficiently dampen liver tumor growth in vivo while the LOXL3-S704D mutant mice were more resistance to single Oxaliplatin treatment than WT mice. Furthermore, we revised the description about treatment schemes, treatment duration, doses, and sample sizes in legends, materials and methods section.

*7. It is surprising that the authors generated CRISPR knock-in mice of *Loxl3-S704D* instead *S704A*. What is the rationale for this? The reviewer appreciates the use of the system and the model, but it is hard to grasp why this mutation is chosen. Indeed, the author's rationale provided in the beginning of the Results section describing this experiment is vague. What is the physiological or pathological relevance of this specific mutation? Furthermore, key comparisons (groups) are missing in this experiment. For example, what is the impact of *S704D* on tumor growth without chemotherapy?*

Response: Thanks for the reviewer's positive appreciation and his or her mention. We are sorry for not describing this experiment so clearly.

We first conformed the loss of function of LOXL3-S704A mutant in xenograft studies (**Fig. 6a-e**). Then, our consideration of the *Loxl3-S704D* mice generation is for the further validation of Oxaliplatin resistant phenotype of LOXL3 in vivo because *Loxl3-S704D* mutant mice with LOXL3 activity similar to that of LOXL3-activated HCC patients, were suitable for testing the combination strategy for HCC patients. Additionally, as the response to the reviewer's previous concern, the Oxaliplatin treatment more resistant *Loxl3-S704D* mutant mice could further be utilized to verify that DHODH was the key contributor to LOXL3-driven chemoresistance in vivo (**Supplemental Fig. S6b-f**). As for the pathological relevance of this specific mutation, we have demonstrated that the phosphorylation level of LOXL3-S704 was positively relative to a worse HCC patient outcome in our **Fig. 7** and **Supplemental Fig. S7**, using a clinical HCC cohort and a PDX model.

Last, the main aim of the in vivo experiments in our study was to confirm the LOXL3 driven Oxaliplatin treatment resistance and the combined therapy efficacy to highlight the importance of our findings and the potential application in human HCC patient treatment, especially the FOLFOX-HAIC regimen we mentioned in our first response. Despite, according to the reviewer's suggestion, we added these comparisons in **Supplemental Fig. S5b**.

8. It is not clear how much of chemo-sensitization is attributed to the mitochondrial function of LOXL3 in a systemic treatment setting. Is it totally independent of its canonical collagen crosslinking function in the tumor microenvironment? Authors mentioned PXS-5153A (a clinically tested LOXL2/3 inhibitor) in the M&M section; however, there is no data shown with this inhibitor. In vivo testing of LOXL2/3 inhibitor in one of the xenograft models in combination with Oxaliplatin and downstream analysis of lipid peroxidation and collagen cross-linking/drug penetration/signaling impact is needed.

Response: Thanks for the reviewer's mention. We initially considered using PXS-5153A to target LOXL3, but this inhibitor was not specific enough to account for inhibiting LOXL3 as PXS-5153A also targeted LOXL2. Though LOXL2 was not so functional in our chemotherapy response screening in vitro assay, the use of this inhibitor PXS-5153A in the systemic treatment experiment could not reliably and specifically confirm the LOXL3's role in chemotherapy resistance in vivo, because we cannot exclude the possibility of LOXL2 expression in other cells in microenvironment to affect the in vivo result of systemic treatment. In addition, we could utilize LOXL3-S704D mice to confirm the LOXL3's role in chemotherapy resistance in vivo by the resistant phenotype of HCC tumors receiving systemic treatment of Oxaliplatin. Thus, we didn't use PXS-5153A eventually, which was deleted in the M&M in the revised manuscript. Hope the specific inhibitor targeting LOXL3 would be developed as soon as possible.

9. The results section can be substantially shortened by removing unnecessary explanations, Furthermore, the rationale of the experiments should be better defined in each section. The discussion section is written like a more "justification" section than the discussion section without citing key studies related to topic of this study. Furthermore, the limitations of the study should be provided.

Response: Thanks for the reviewer's constructive suggestion. We removed some explanations in the results section. And we rewrote the introduction and discussion. At this occasion, the limitations of the study were provided in the discussion section.

Minor points:

1. Fig. 11 is not mentioned in the text.

Response: Thanks, we supplemented the description for Fig. 11.

2. Western blot images need to have the molecular weight of the proteins shown next to the images.

Response: Thanks for the reviewer's mention, we organized our western blot images and all marked the molecular weight of the proteins in our original data of western blot. Please check it. We apologize for not marking the molecular weight of the proteins in Fig., for saving the space of Fig. in our manuscript, which is already too full.

3. It is not clear if 5-FU or Oxaliplatin is used in Fig. S1L.

Response: Thanks for the reviewer's mention, we correct this mistake. It should be Oxaliplatin.

4. The impact of stable knockdown of LOXL3 on the other LOX family members need to be shown at mRNA and protein levels.

Response: Thanks for the reviewer's mention, we supplemented the data in Fig. S1E. The same reason for saving the space of Fig. in our manuscript, we apologize again for not showing the mRNA level in this study, just the protein level. We measured the mRNA level of LOXL3 on the other LOX family members, there is truly no difference, same as the protein level. We

supplement the mRNA data here for checking by the reviewer.
The result is following:

5. *Statistical tests and comparison groups needs to be clearly stated.*

Response: Thanks for the reviewer's mention, we have added lines to link the important cell groups needed to be highlighted and compared, for drawing the conclusion. We hope that it is now clearer to examine our comparisons for important groups.

6. *The details of the PDX models used need be provided.*

Response: Thank a lot for the reviewer's positive comment. According to the reviewer's suggested, we supplemented the clinic-pathological information of the patients offering the tumor tissues for establishing the PDX model. The process of our PDX models have been describe in detail into our method in the part of experimental mice.

REVIEWER COMMENTS

Reviewer #1 (Remarks to the Author):

The authors have adequately addressed the questions from this reviewer. The manuscript has been significantly improved and can be accepted for its publication in Nature Communications.

One point, the uridine supplementation experiment (Supplementary Fig. S3i-k; to address an insightful question from reviewer 3) is very nice and adds additional support to their conclusion that DHODH's function in regulating mitochondrial lipid peroxidation and ferroptosis, but not its canonical function in regulating pyrimidine biosynthesis, is important for its biological effect studied in this context. To guide readers better, I suggest the authors to provide a bit more information here: DHODH has been shown to suppress mitochondrial lipid peroxidation and ferroptosis through generating ubiquinol (a radical trapping antioxidant) but independent of its ability in synthesizing pyrimidine. Therefore, uridine supplementation can be used as an approach to separate its function in pyrimidine biosynthesis and ferroptosis defense (and cite relevant publication).

Reviewer #2 (Remarks to the Author):

This is the revised version of the manuscript. The authors experimentally addressed numbers of the raised concerns and substantiated their findings. The revision significantly strengthened the manuscript.

comments:

It would be beneficial to clearly point out that the suggested treatment approach/treatment, i.e. FOLFOLX, is not a global strategy for HCC but rather used in selected patients.

Reviewer #3 (Remarks to the Author):

The authors have addressed some of my concerns adequately; however, several key points are still not addressed sufficiently.

It is critical to know the (oxaliplatin and 5-FU) resistance status of the two major cell lines used throughout the study if the study claims to address chemoresistance in liver cancer (this reviewer is not asking the definitions of acquired or de novo resistance).

Another point that needs to be addressed is the EGFR activation upon oxaliplatin or 5-FU as this is another major starting point in this manuscript.

The authors also did not address the comment related to the effect of DHODH downregulation on mitochondrial respiratory chain although they now show that mitochondrial ROS is substantially induced upon LOXL3 inhibition under oxaliplatin treatment (figure for reviewer). If DHODH is in the center of the proposed mechanism, it is important to know which DHODH function(s) is critical in chemoresistance.

Importantly, it is not clear why authors do not target LOXL3. Instead, they target its downstream DHODH1 whose level does not change even in LOXL3 wt vs mutant tumors (as Reviewer 1 also indicated) in combination with chemotherapy to prove the role of LOXL3 in chemoresistance in liver cancer which might have translational potential. The presented in vivo experiments with DHODH1 in combination with oxaliplatin are lacking the LOXL3 context. It is understandable that there is no LOXL3 specific inhibitor; however, there are LOXL2/LOXL3 inhibitors as authors mentioned, and LOXL2 is not localized in mitochondria, and not involved in chemoresistance in liver cancer according to the authors' findings, justifying the use of these inhibitors.

Finally, authors did not address (answer) the comment on the total exclusion of canonical function

of LOXL3 in chemoresistance in this paper. This needs to be addressed experimentally.

RESPONSE TO REVIEWERS' COMMENTS

Reviewer comments:

Reviewer #1:

The authors have adequately addressed the questions from this reviewer. The manuscript has been significantly improved and can be accepted for its publication in Nature Communications. One point, the uridine supplementation experiment (Supplementary Fig. S3i–k; to address an insightful question from reviewer 3) is very nice and adds additional support to their conclusion that DHODH's function in regulating mitochondrial lipid peroxidation and ferroptosis, but not its canonical function in regulating pyrimidine biosynthesis, is important for its biological effect studied in this context. To guide readers better, I suggest the authors to provide a bit more information here: DHODH has been shown to suppress mitochondrial lipid peroxidation and ferroptosis through generating ubiquinol (a radical trapping antioxidant) but independent of its ability in synthesizing pyrimidine. Therefore, uridine supplementation can be used as an approach to separate its function in pyrimidine biosynthesis and ferroptosis defense (and cite relevant publication).

Response: We thank a lot for the reviewer's constructive suggestion. As such, we cited relevant publications to provide more information about the role of DHODH in lipid peroxidation and ferroptosis by the ubiquinol generation but independent of its ability in synthesizing pyrimidine.

Reviewer 2#:

This is the revised version of the manuscript. The authors experimentally addressed numbers of the raised concerns and substantiated their findings. The revision significantly strengthened the manuscript.

Comments:

It would be beneficial to clearly point out that the suggested treatment approach/treatment, i.e. FOLFOX, is not a global strategy for HCC but rather used in selected patients.

Response: We thank a lot for the reviewer's constructive suggestion. As the reviewer suggested, we clearly pointed out in our revised manuscript that the suggested treatment approach/treatment, i.e., FOLFOX, is not a global strategy for HCC but rather used in the selected patients.

Reviewer 3#:

The authors have addressed some of my concerns adequately; however, several key points are still not addressed sufficiently.

Response: We apologized for not fully addressing the concerns of the reviewer in the first round revision. Here, we collated the background knowledge and integrated it with our research, and articulated it as clearly as possible, to address the reviewer's concerns.

1. It is critical to know the (oxaliplatin and 5-FU) resistance status of the two major cell lines used throughout the study if the study claims to address chemoresistance in liver cancer (this reviewer is not asking the definitions of acquired or de novo resistance).

Response: Thanks for the reviewer's mention. In order to know the resistance status of the two major cell lines (Huh7 and Hep3B), we measured IC₅₀ (half maximal inhibitory concentration) by drug response curve, which implied the ability of cells to resist drug treatment. The lower IC₅₀ value means the lower drug resistance or higher drug sensitivity. Thus, it was observed that, under oxaliplatin treatment, shLOXL3 cells exhibited much lower IC₅₀ value compared to control cells, indicating the reduced chemoresistance status and increased chemosensitivity status of Huh7 and Hep3B cells (**new Supplementary Fig. S1e**).

2. Another point that needs to be addressed is the EGFR activation upon oxaliplatin or 5-FU as this is another major starting point in this manuscript.

Response: We are sorry for not illustrating this concern clearly for the reviewer in the last revision. Actually, in **Supplementary Fig. S2n**, it was proved that cells treated with oxaliplatin increased the phosphorylation level of EGFR and activated EGFR, offering **the direct evidence for EGFR activation upon oxaliplatin treatment**. Consistently, compared with only oxaliplatin treatment, combination with cetuximab, an EGFR inhibitor, significantly increased the cell death (**Supplementary Fig. S2a**), offering another evidence for EGFR activation upon oxaliplatin treatment leading to increased chemoresistance ability, which was consistent with previous reports (**PMID: 24295852, PMID: 22157681, PMID: 23242808, PMID: 21741919**). We have added these references in the revised manuscript.

3. The authors also did not address the comment related to the effect of DHODH downregulation on mitochondrial respiratory chain although they now show that mitochondrial ROS is substantially induced upon LOXL3 inhibition under oxaliplatin treatment (figure for reviewer). If DHODH is in the center of the proposed mechanism, it is important to know which DHODH function(s) is critical in chemoresistance.

Response: Thanks for the reviewer's mention and we feel sorry for not addressing this concern clearly for the reviewer in the last revision. We agree with the reviewer that it is important to know which DHODH function is critical in chemoresistance of liver cancer.

Dihydroorotate dehydrogenase (DHODH) is a mitochondrial enzyme to catalyze oxidize dihydroorotate (DHO) to orotate (OA) by removing two electrons from DHO using its redox-active flavin mononucleotide (FMN) prosthetic group. Ubiquinone (CoQ) acts as an electron acceptor that transports electrons obtained from DHODH to mitochondrial respiratory complex III, disposing of the electrons at FMN to complete the catalytic cycle and forming CoQH₂. Therefore, CoQ/CoQH₂ links DHODH to mitochondrial respiratory chain (**PMID: 28666740**). The mitochondrial respiratory chain is an electron transport system composed of four main enzyme complexes (I, II, III and IV) located in the inner membrane of mitochondria (**PMID: 19026783**). DHODH needs to deposit electrons onto CoQ and change CoQ into CoQH₂, which

is then re-oxidized by means of mitochondrial respiratory chain CIII/IV-mediated respiration. Meanwhile, as CoQH₂ is a lipophilic radical-trapping antioxidant to detoxify lipid peroxyl radicals, a previous report, by Mao and Professor Gan, proved that DHODH regulates ferroptosis through reducing CoQ to CoQH₂ in mitochondria, which leads to the combination of ferroptosis regulation by DHODH with its mitochondrial respiratory chain regulation role (**PMID: 33981038**).

Briefly, DHODH catalyzes the conversion of DHO to OA in the pyrimidine biosynthesis pathway. In the catalytic process, DHODH removes two electrons from DHO to mitochondrial respiratory complex via CoQ, leading to the reduction (a biochemical usage, derived from reduction reaction) of CoQ to CoQH₂ in the mitochondrial inner membrane which resulted in the tight combination of the effect of DHODH on mitochondrial respiratory chain with ferroptosis regulation. In **Supplementary Fig. S3i-k**, we measured uridine level and found the supplementation of uridine did not enhance oxaliplatin resistance of HCC LOXL3-S704A cells, which implied that enhanced liver cancer chemoresistance by LOXL3 was independent of pyrimidine biosynthesis function of DHODH. Therefore, it could be concluded that DHODH mediated the effect of LOXL3 on ferroptosis and liver cancer chemoresistance, depending on the its function on reducing CoQ to CoQH₂ in mitochondrial respiratory chain.

For further confirmation by experiments, we supplemented mitochondria-targeted analogues of CoQ and CoQH₂ to LOXL3-S704A mutant cells under oxaliplatin treatment. The result showed that CoQH₂, but not CoQ, could protect the lipid peroxidation and cell death induced by DHODH deficiency in LOXL3-S704A mutant cells under oxaliplatin treatment (**new Supplementary Fig. S3l-m**). Furthermore, as electron transport chain (ETC) complex III converts CoQH₂ back into CoQ, we treated cells with the complex III inhibitor antimycin A (Anti A) to decrease the CoQ/CoQH₂ ratio. Similarly, same with the mitoQH₂ supplementation, antimycin A protected cells against oxaliplatin treatment-induced ferroptosis (**new Supplementary Fig. S3n-o**). Together, our data offered the experimental evidence to support the finding that LOXL3 regulated DHODH protein stability to inhibit ferroptosis by reducing CoQ to CoQH₂ in mitochondria under oxaliplatin treatment, resulting in increased chemoresistance.

As for the effect of DHODH downregulation on the overall mitochondrial respiratory chain, it could be detected by measuring oxygen consumption and ATP production level. A previous report revealed that there was no more than 5%-10% of routine oxygen consumption and ATP production within OXPHOS in DHODH deficiency cells, suggesting that direct contribution of DHODH to overall ATP generation is only at baseline and DHODH knockout cells feature normal levels of OXPHOS-derived ATP and bioenergetics (**PMID: 30449682**). For further confirmation by experiments, we measured the level of oxygen consumption rate and ATP production in LOXL3-S704 mutant Huh7 or Hep3B cells that lacking of DHODH expression, the result indeed showed a slight decrease (**new Supplementary Fig. S3p-q**), which may be explained that CoQ serves as an electron acceptor both for DHODH and the other upstream ETC complexes specific in liver cancer cells, leading to the overall balance of mitochondrial respiratory chain.

4. Importantly, it is not clear why authors do not target LOXL3. Instead, they target its downstream DHODH1 whose level does not change even in LOXL3 wt vs mutant tumors (as

Reviewer 1 also indicated) in combination with chemotherapy to prove the role of LOXL3 in chemoresistance in liver cancer which might have translational potential. The presented in vivo experiments with DHODH1 in combination with oxaliplatin are lacking the LOXL3 context. It is understandable that there is no LOXL3 specific inhibitor; however, there are LOXL2/LOXL3 inhibitors as authors mentioned, and LOXL2 is not localized in mitochondria, and not involved in chemoresistance in liver cancer according to the authors' findings, justifying the use of these inhibitors.

Response: Thanks for the reviewer's concern. The reason why we did not target LOXL3 in vivo just because there is no LOXL3 specific inhibitor at present. For our translational aim to reduce chemotherapy resistance and drug toxicity, we chose to use DHODH1 inhibitor in combination with oxaliplatin in *in vivo* experiments, considering that DHODH inhibitors have already been approved by the FDA or extensively tested in the clinic.

Moreover, though LOXL2 is not localized in mitochondria, and not involved in chemoresistance in liver cancer, we could not exclude the possibility of LOXL2 to involve in other factors affecting tumor, like tumor metastasis or tumor microenvironment in vivo, which would lead to further exclusion and confirmation if the result of LOXL2/LOXL3 inhibitor in vivo is not so satisfied.

As for the DHODH protein level does not change in LOXL3-WT vs mutant tumors, first, we actually showed the difference of DHODH protein level in the liver of LOXL3-WT and mutant mice **before** sleeping beauty induction in liver tumorigenesis. Secondly, a reasonable and understandable explanation is that: it is a process that cells abnormally response to signals and stress which eventually results in tumor progression and resistance to chemotherapy. Our data showed that DHODH protein level would be elevated in response to oxaliplatin treatment for defending the stress. Although DHODH expression level of WT mice caught up with *Loxl3-S704D* mice after treatment with oxaliplatin at 6th week, the degree of response to oxaliplatin was different from the beginning of the front because at the beginning, the liver from *Loxl3-S704D* mice contains more DHODH which conferred the *Loxl3-S704D* mice with more resistance to oxaliplatin induced lipid peroxidation. Once the HCC cells response to the oxaliplatin treatment, after accumulation for 6 weeks, the liver from *Loxl3-S704D* mice were much more resistant to oxaliplatin induced lipid peroxidation than wild type, leading to the tumor phenotype between WT and S704D mice under oxaliplatin treatment.

The last we have to point out that the presented data of in vivo experiments with DHODH in combination with oxaliplatin are not lacking the LOXL3 context. Actually, we utilized *Loxl3-S704D* mice to confirm the LOXL3's role in chemotherapy resistance in vivo by the resistant phenotype of HCC tumors receiving systemic treatment of oxaliplatin. In **supplementary Fig. S6b-f**, it was showed that **DHODH inhibitor could block the liver cancer chemoresistance phenotype of *Loxl3-S704D* mice, which indicated the essential role of DHODH in LOXL3 mediated liver cancer chemoresistance in vivo.**

5. Finally, authors did not address (answer) the comment on the total exclusion of canonical function of LOXL3 in chemoresistance in this paper. This needs to be addressed experimentally.

Response: Thanks for the reviewer's mention and we apologize for not addressing the

comment clearly. As LOXL3-K35/K36A has major LOXL3 activity in matrix (**new Supplementary Fig. S2g-h**), but lacks of the ability to enter into mitochondrial by losing the binding with TOM20 (**Fig. 2f-g, Supplementary Fig. S2d-e**), we used LOXL3-K35/K36A to exclude the canonical function of LOXL3 in chemoresistance of liver cancer. Firstly, we restored LOXL3 expression of WT or K35/K36A mutant in LOXL3 knockdown HCC cells (**supplementary Fig. S2f**). Next, it was observed that LOXL3-WT, but not LOXL3-K35/K36A mutant, could rescue the lipid peroxidation level (**Fig. 2h-i**), cell death, drug response status (**new Supplementary Fig. S2i-j**) and DHODH protein level (**Fig. 2j**) under the treatment of oxaliplatin, which could exclude the canonical function of LOXL3 in chemoresistance of liver cancer.

REVIEWERS' COMMENTS

Reviewer #3 (Remarks to the Author):

Most of my concerns have been addressed adequately

RESPONSE TO REVIEWERS' COMMENTS

Reviewer #3 (Remarks to the Author):

Most of my concerns have been addressed adequately.

Response: Thanks for your comments and your patience for improving this study.